



**Orbitally forced environmental changes during the accumulation**
**of a Pliensbachian (Lower Jurassic) black shale in northern Iberia**
Naroa Martinez-Braceras[1,2]; Aitor Payros[1]; Jaume Dinarès-Turell[3]; Idoia Rosales[4]; Javier
Arostegi[1] and Roi Silva-Casal[5]
[1] Department of Geology, Faculty of Science and Technology, University of the Basque
Country (UPV/EHU), P.O. Box 644, 48080 Bilbao, Spain
[2] Laboratorio de Evolución Humana, Departamento de Historia, Geografía y
Comunicación, Universidad de Burgos, Edificio I+D+I, Plaza de Misael Bañuelos/n,
09001 Burgos, Spain
[3] Istituto Nazionale di Geofisica e Vulcanologia, Via di Vigna Murata 605, 00142 Rome,
Italy
[4] Centro Nacional Instituto Geológico y Minero de España (IGME, CSIC), La Calera 1,
Tres Cantos, 28760 Madrid, Spain
[5] Dpto. Dinàmica de la Terra i de l'Oceà, Facultat de Ciències de la Terra, UB. 08028
Barcelona.
Correspondence to: Naroa Martínez Braceras (naroa.martinez@ehu.eus)
**Abstract**
Lower Pliensbachian hemipelagic successions from the north Iberian palaeomargin are
characterized by the occurrence of organic-rich calcareous rhythmites of decimetre-thick
limestone and marl beds and thicker black shale intervals. Understanding the genetic
mechanisms of the cyclic lithologies and involved processes along with the nature of the
carbon cycle is of primary interest. The cyclostratigraphic study carried out in one of the
black shales exposed in Santiurde de Reinosa (Basque-Cantabrian Basin) revealed that
the calcareous rhythmites responded to periodic environmental variations in the
Milankovitch-cycle band, with the prevalence of precession, short eccentricity and long
eccentricity cycles.
The main environmental processes that determined the formation of the rhythmite were
deduced on the basis of the integrated sedimentological, mineralogical and geochemical
study of an eccentricity bundle. The formation of precession couplets was controlled by
variations in carbonate production and dilution by terrigenous supplies, along with
periodic changes in bottom water oxygenation. Precessional configurations with marked
annual seasonality, increased terrigenous input (by rivers or wind) to marine areas and
boosted organic productivity in surface waters. The great accumulation of organic matter
on the seabed eventually decreased bottom waters oxygenation, which might also be
influenced by reduced ocean ventilation. Thus, deposition of organic-rich marls and
shales occurred when annual seasonality was maximum. On the contrary, a reduction in
terrestrial inputs at precessional configurations with minimal seasonality disminshed



shallow organic productivity, added to an intensification of vertical seawater mixing,
contributed to increasing the oxidation of organic matter. These conditions also favoured
greater production and basinward exportation of carbonate mud in shallow marine areas,
causing the formation of limy hemipelagic beds. Short eccentricity cycles modulated the
amplitude of precession driven variations in terrigenous input and oxygenation of bottom
seawaters. Thus, the amplitude of the contrast between successive precessional beds
increased when the Earth's orbit was elliptical and diminished when it was circular. The
data also suggest that short eccentricity cycles affected short-term sea level changes,
probably through orbital modulated aquifer-eustasy.

**1. Introduction**

As a consequence of the gravitational interaction between astronomical bodies, the
Earth's axial orientation and orbit vary cyclically at timescales that range from tens of
thousands to few million years (Berger and Loutre, 1994). These variations in orbital
configuration regulate the latitudinal distribution of solar radiation (insolation), which
determines the contrast between seasons. These periodic changes in the climatic system
can be recorded as cyclic stratigraphic successions, the so-called rhythmites, in a wide
range of sedimentary environments (Einsele and Ricken, 1991). As the open ocean is
hardly affected by processes that may interrupt the continuous settling of fine-grained
particles or erode the seabed, deep marine pelagic sediments accumulate at a generally
constant, but slow (few cm/ky), sedimentation rate. Thus, pelagic rhythmites from both
oceanic sediment cores and indurate successions contain accurate records of orbitally
modulated, quasi-periodic climate-change episodes, which provide high-resolution
astrochronologies (Hinnov, 2013).
Significant progress in Early Jurassic cyclostratigraphy has been made in the last few
decades thanks to the study of exceptional orbitally modulated sedimentary records
obtained from deep marine environments of the Perytethyan realm (e.g., Cardigan and
Cleveland Basins by Hüsing et al., 2014, Storm et al., 2020; Pieńkowski et al., 2021; Paris
Basin by Charbonnier et al., 2023). Although these studies provided relevant
astrochronological information, they did not focus on the climatic and environmental
impact of the orbital cycles on the sedimentary record. Other studies deduced a control of
long-term orbital cycles on the Jurassic carbon cycle (Martinez and Dera, 2015; Ikeda et
al., 2016; Hollar et al 2021; Zhang et al., 2023), but the climatic and environmental
influence of short-term cycles has been less studied (Hinnov and Park, 1999; Ikeda et al.,
2016; Hollar et al., 2023).
The aim of this study was to analyze the climatic and environmental impact of short-term
astronomical cycles on Lower Jurassic deep marine deposits. To this end, a hemipelagic
alternation of limy and marl/shale beds was analyzed in the Santiurde de Reinosa section
(hereafter referred to as the Santiurde section), Basque-Cantabrian Basin (BCB),
Cantabria province, Spain. In order to determine if sedimentation was orbitally forced, a
cyclostratigraphic analysis of the hemipelagic rythmites was firstly undertaken.
Subsequently, an integrated multiproxy study was performed in a selected interval of the





section in order to disentangle what sedimentary processes and environmental factors influenced on the formation of the hemipelagic rhytmites.

## 2. Geological setting

In early Jurassic times the BCB was located to the south of Armorican and to the north of the Iberian Massif, being part of the Laurasian epicontinental seaway which connected the Boreal Sea with the southern Tethyan Ocean (Fig. 1A; Aurell et al., 2002; Rosales et al., 2004). Previous palaeogeographic reconstructions located the north Iberian margin at approximately 30ºN palaeolatitude (Quesada et al., 2005; Osete et al., 2010). Hence, the source area was located in the semiarid belt but close to the boundary with the humid climatic zone (temperate climate characterized by megamonsoons; Dera et al., 2009; Deconinck et al., 2020), which made it especially sensitive to astronomically driven climate change.

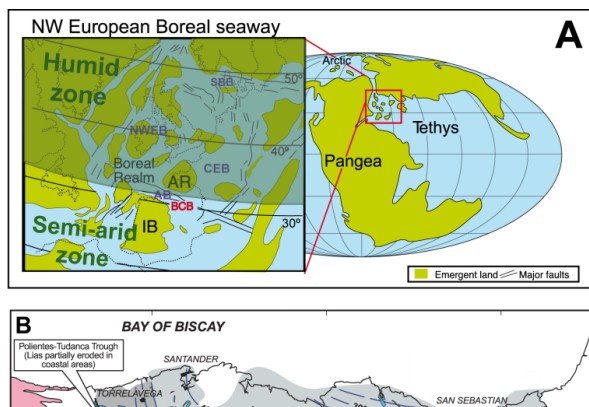

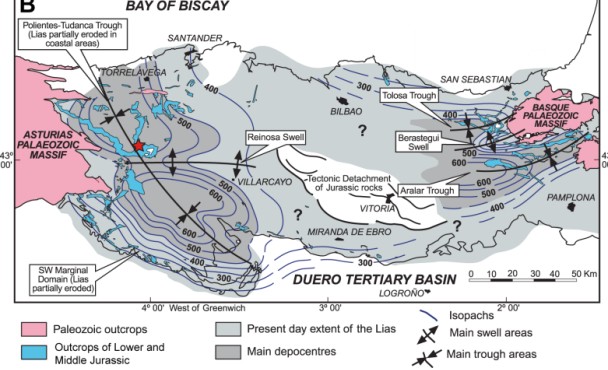

Figure 1. A) Palaeogeography and climatic zonation (modified from Quesada et al., 2005; Dera et al., 2009; Ostete et al., 2010) of Western Europe in Early Juassic times. IB: Iberian massif, AR: Armorican massif, AB: Asturian Basin, BCB: Basque-Cantabrian Basin, CEB: Central European Basin, NWEB: NW European Basin, SBB: South Boreal Basin. B) Simplified geographic and geological map of lower and middle Jurassic outcrops in the BCB area, with location of the studied Santiurde section (red star). Superimposed isopach map shows the thickness of the lower Jurassic rocks and the basin configuration in sedimentary troughs and swells (modified from Quesada et al., 2005).

Hettangian and lower Sinemurian deposits accumulated in evaporitic tidal flats and shallow carbonate ramps, whereas the overlying Sinemurian-Callovian succession was accumulated in relatively deep, open marine conditions (Aurell et al., 2002; Quesada et al., 2005). Differential subsidence during the Jurassic related to early mobilization of underlying Triassic salt resulted in the creation of several troughs in the BCB (Fig. 1B, Quesada et al., 2005).



Pliensbachian (192.9–184.2 Ma) hemipelagic successions of the BCB (Camino
Formation; Quesada et al., 2005) are characterized by the occurrence of three black shale
(BS) intervals, each several tens of metres thick (Braga et al., 1988; Quesada et al., 1997,
2005; Quesada and Robles 2012; Rosales et al., 2001, 2004, 2006). These BS intervals
can be correlated with similar coeval deposits in neighbouring basins in Asturias (Borrego
et al., 1996; Armendáriz et al., 2012; Bádenas et al., 2012, 2013; Gómez et al., 2016).
Coeval organic rich marine facies have also been observed in other Thetyan lower
Jurassic successions from Portugal (Silva et al., 2011), United Kindom (Hüsing et al.,
2014), France (Bougeault et al., 2017) and Germany (Pieńkowski et al., 2008). The BCB
Pliensbachian BS intervals present relatively high organic carbon content (2–6wt%), high
pyrite concentrations and scarce benthic faunas. Thermal maturity analysis showed that
the BS intervals found at the depocentres are overmature today, but they sourced the only
oil reservoir discovered inland Iberia (Quesada et al., 1997, 2005; Quesada and Robles,
2012; Permanyer et al., 2013). Pyrolysis of thermally immature samples from marginal
areas showed total organic carbon values of up to 20 wt% and hydrogen index values up
to 600-750 mg HC/g TOC (Suárez-Ruiz and Prado, 1987; Quesada et al., 1997). Analyses
on organic matter (OM) showed that the assemblage is mainly composed of marine type-
II kerogens, in which amorphous and algal material prevail (Quesada et al., 1997, 2005;
Permanyer et al., 2013). More specifically, the analysis revealed a low content in
gammaceranes, which suggests normal salinity conditions, and great abundance of
triclinic triterpanes, which can be associated to *Tasmanites* type unicellular green algae
with organic theca. In addition, the high content in isorenieratene byproducts, such as
aryl-isoprenoids, indicates the occurrence of photosynthetic and sulfurous green algae
communities (*Chlorobiaceae*) developed in oxygen-depleted conditions.
The Santiurde section studied herein is exposed at exit 144 of motorway A67 (UTM
X411431.091 Y4769002.593; Fig. 1B), approximately 50 km south-west of Santander
and 1 km north-west of a coeval section studied by others at the train station in the same
locality (e.g. Rosales et al., 2001, 2004, 2006; Quesada et al., 2005), with which a bed-
by-bed correlation can be readily carried out. The studied succession begins with 2.5 m
of alternating grey limestones and thin marlstones (Puerto Pozazal Formation), which are
overlain by 20 m of the lower part of the Pliensbachian Camino Formation, mainly made
up of alternations of hemipelagic marls, limestones and overmature black shales (Rosales
et al., 2004; Quesada et al., 2005). The section includes the oldest (*Uptonia jamesoni*
Zone) of the BS intervals identified in the Camino Formation (Fig. 2A).

## 3. Materials and methods

### 3.1. Cyclostratigraphic analysis of the Santiurde section

A detailed cm-scale stratigraphic log was measured in a 30.40 m thick succession that
exposes the transition from the Puerto Pozazal Formation to the Pliensbachian Camino
Formation. A broad range of sedimentological features, such as bed shape, thickness,
composition, palaeontological content and structures, were annotated. A total of 373 hand
samples were collected, with a resolution of at least 3 samples per bed, avoiding visible



skeletal components, burrows and veins. The weight normalized low-field magnetic susceptibility (MS) of the samples was measured using a Kappabridge KLY-3 instrument (Geophysika Brno) housed at the Geology department of the University of the Basque Country, Bilbao, Spain. Subsequently, rock-powder samples were obtained and stored in transparent antiglare prismatic vials, which were scanned in a dark room using a desktop office scanner. The average colour (RGB value) of the scanned images of rock-powder samples was determined using the ImageJ software and following the protocol in Dinarès-Turell et al. (2018) and Martinez-Braceras et al. (2023).

In order to carry out a cyclostratigraphic analysis, the Acycle software (Li et al., 2019) and the Astrochron package for R (Meyers et al., 2014) were used. The MS and colour data series were linearly interpolated and detrended first. Subsequently, power spectra were obtained using the $2\pi$-Multi Taper Method (MTM) with three tapers, and confidence levels (CL) were calculated following robust red-noise modelling (Mann and Lees 1996). In addition, Evolutive Harmonic Analysis (EHA; Meyers et al., 2001) and Wavelet analyses (Torrence and Compo, 1998) were also carried out in order to examine the variability of the main frequency bands throughout the succession. Finally, the most significant frequency bands identified in the data series were isolated by Gaussian bandpass filtering.

### 3.2. Multiproxy analysis of Bundle 9

An integrated analysis of several environmentally sensitive proxies was undertaken in the 19 beds found between 20.30 and 23.85 m of the stratigraphic succession. This interval includes a complete eccentricity bundle (B9 see results below), as well as the uppermost and lowermost beds of the underlying and overlying bundles, respectively. Fifty-seven samples were collected to perform a calcimetric analysis by measuring the carbonate percentage in 1 g of powder of each sample using a FOGL digital calcimeter (BD inventions; accuracy of 0.5%) housed at the University of the Basque Country. These samples were also analysed for inorganic $\delta^{13}C_{carb}$ and $\delta^{18}O_{carb}$ content at the Leibniz Laboratory for Radiometric Dating and Stable Isotope Research (Kiel University, Germany) using a Kiel IV carbonate preparation device connected to a ThermoScientific MAT 253 mass spectrometer. Precision of all internal and external standards (NBS19 and IAEA-603) was better than ±0.05‰ for $\delta^{13}C_{carb}$ and ±0.09‰ for $\delta18O_{carb}$. All values are reported in the VPDB notation relative to NBS19.

In addition, one sample from the central part of each bed was studied for petrographic and scanning electron microscope (SEM) analysis, mineralogical content, elemental composition and organic geochemistry. For the mineralogical and geochemical analyses, the samples were ground in the laboratory. Whole-rock mineralogy was obtained by analysing randomly oriented rock powder by X-ray diffraction (XRD), using a Philips PW1710 diffractometer (Malvern Panalytical, Malvern, UK) at the University of the Basque Country. The step size was 0.02° 2h with a counting time of 0.5 s per step. Major and trace element concentrations were determined at the University of the Basque Country using a Perkin-Elmer Optima 8300 spectrometer (ICP-OES; PerkinElmer) and a



Thermo XSeries 2 quadrupole inductively coupled plasma mass spectrometer (ICP-MS;
Thermo Fisher Scientific) equipped with a collision cell, an interphase specific for
elevated total dissolved solids (Xt cones), a shielded torch, and a gas dilution system.
Analysis of the JG-2 granite standard and error estimates of each element showed that the
uncertainty of the results corresponds to the 95% confidence level. Finally, organic
carbon ($C_{org}$) and organic nitrogen ($N_{org}$) contents, as well as their isotopic $\delta^{13}C_{org}$ and
$\delta^{15}N_{org}$ values were obtained by combustion of powdered and decarbonated samples in
an elemental analyzer Flash EA 1112 (ThermoFinnigan) connected to a DeltaV
Advantage mass spectrometer (Thermo Scientific) at the University of A Coruña, Spain.
Calibration of $^{13}C_{org}$ and $^{15}N_{org}$ was done against certificated standards USGS 40,
USGS41a, NBS 22 and USGS24. Results are expressed in the VPDB notation, accuracy
(standard deviation) being ±0.15‰.
In order to explore compositional relationships and trends using comprehensive multi-
elemental datasets, Pearson correlation coefficients (r) and their significance (P-values)
were estimated for pairs of variables using the SPSS 28 statistical package (IBM
Corporation, SPSS statistics for Windows, version 28.0.1.1, 2022, Armonk, NY, USA).
In addition, a multivariate factor analysis was undertaken with the aim of identifying the
number of virtual variables (factors) that explains the highest percentage of the variability
in the analyzed dataset.

## 205   4. Results

### 206   4.1. General Santiurde section

### 207   4.1.1. Sedimentology and petrography

The succession displays an alternation of weather resistant, light coloured, bioturbated
limestones or marly limestones, and weather recessive, dark coloured, laminated marls or
shales (Fig. 2). In the outcrop, the fossil record of limestones and marly limestones is
dominated by isolated ammonites, belemnites and brachiopods (Fig. 2C). Thin sections
show mudstones and wackestones with dispersed benthic foraminifera, fragmented
echinoderms, brachiopods and pyritized bivalve shells (mainly pectinids) in a microspar
matrix (Figs. 3A and C). Well-preserved placoliths of coccolithophorids were identified
by SEM (Figs. 3C and G).
Both marls and shales constitute friable, weather-recessive beds, the latter generally
showing darker colour and more prominent lamination (Fig. 2D), also observed in thin
sections (Fig. 3B). Furthermore, marls contain nekto-planktonic fossils (ammonites,
belemnite and calcareous unicellular algae) and evidence of benthonic communities
(pyritized shells of bivalves and rhynchonellid brachiopods; trace fossils, such as
*Chondrites* and *Planolites*), whereas the latter are absent in shales. This is confirmed by
SEM analysis, as marls contain isolated, broken and randomly oriented clay minerals that
wrap well-preserved coccoliths and calcispheres with signs of bioturbation (Fig. 3C, 3D
and 3G). Nektonic organisms and placktonic unicellular algae also occur in shales, but
benthonic fauna and bioturbation are virtually absent. SEM observations also showed that



the lamination in shales is caused by the alternation of detrital components (mainly clays
but also quartz) and organic components (such as bitumen, polymeric extracellular
substances linked to biofims, filamentous bacterial mats, or fungal hyphae; Fig. 3E and
3F). Pyrite fambroids are more common in shales than in limy beds (Fig. 3H).

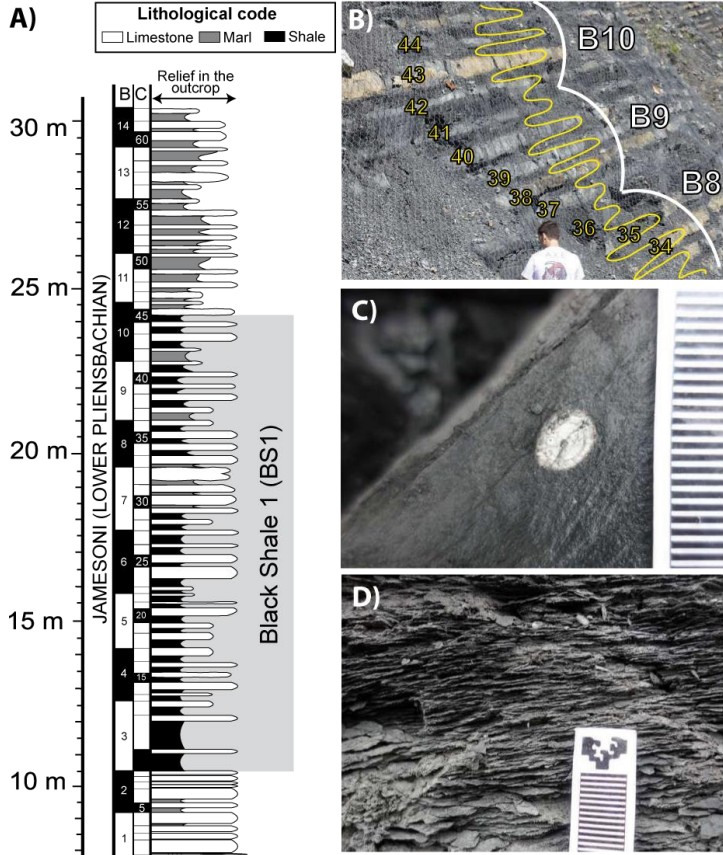

Figure 2. A) Synthetic lithological log of the Santiurde section, including chronostratigraphy from Quesada et al. (2005)
and Rosales et al. (2006). B) Calcareous couplets (yellow numbers) of bundles 8 to 10 (white numbers) in the Santiurde
outcrop. The white curve shows the relief of successive beds in the outcrop (left, weather recessive; right, weather
resistant), which is mainly determined by their carbonate content. C) Close up of a marly limestone with a partly
pyritized belemnite. D) Close up of a laminated black shale. Scale bar in mm.
A total of 62 calcareous couplets (C1 to C62) were identified in the studied succession,
whose thicknesses vary from 8 to 97 cm, averaging out at 36 cm (Fig. 2A and 4). These
couplets display a larger-scale arrangement in 12 complete bundles plus another two
incomplete bundles at the base and top of the section. These bundles range in thickness
from 126 to 208 cm (average: 167.3 cm) and are composed of four to six couplets
(generally five). Bundles typically contain a package of three prominent central couplets
(with significant lithological contrast between successive limestone and marl/shale beds)
which is underlain and overlain by less obvious couplets (lower lithological contrast
between successive marl and marly limestone beds).

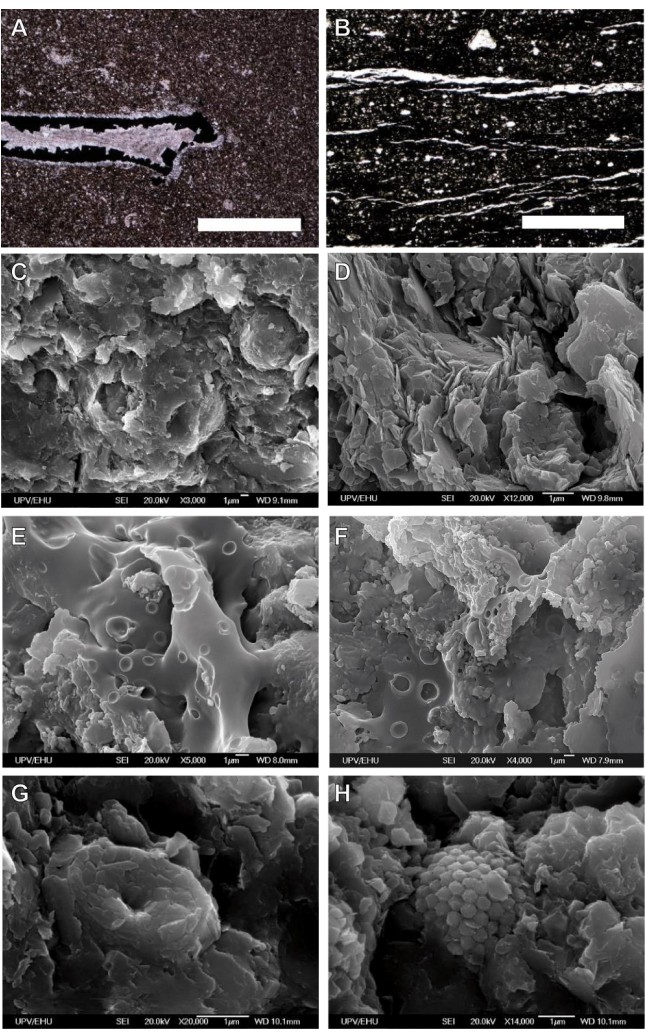

Figure 3. Petrographic views of limestone C41 (A) and shale C36 (B). The white bars represent 1 mm. C) General
texture of a limestone bed (couplet C37), showing partly dissolved and broken coccoliths and calcispheres. D) General
texture of a marly bed (couplet C37) with evidence of bioturbation. E) and F) probable biofilms. F) Well preserved
coccolith. G) Pyrite fambroid.

## 4.1.2. Colour, calcium carbonate and magnetic susceptibility

Colour values (mean RGB) range from 69.87 to 158.99, averaging out at 102.73 (Fig. S1;
Table S1). The colour curve oscillates in line with the lithological alternation, colour
values generally being higher in limestones and marly limestones (average of 115.14)
than in intervening marls or shales (average of 90.71). The variations in colour values are
more significant in the central couplets of bundles than at bundle boundaries. This
suggests that, as shown in previous studies (Dinarès-Turell et al., 2018; Martínez-
Braceras et al., 2023), colour values are representative of the carbonate content of the
samples. This is confirmed by the carbonate content analysis carried out between couplets





C35 to C44 (see below), as both colour and carbonate content show the same arrangement
in couplets and bundles (r: 0.89, p<0.001; Fig. S1).
Weight-normalized magnetic susceptibility values range from $5.08 \times 10^{-06}$ to $1.67 \times 10^{-06}$
$m^3/kg$, averaging out at $9.9 \times 10^{-06}$ $m^3/kg$ (Fig. S1, Table S1). In most cases, limestones
and marly limestones have higher susceptibility (average: $1.08 \times 10^{-05}$ $m^3/kg$) than shales
and marls (average: $8.99 \times 10^{-06}$ $m^3/kg$). The MS of hemipelagic deposits is commonly
determined by their paramagnetic components (mostly detrital clays; Kodama and
Hinnov, 2015). However, in Santiurde this parameter does not show a great correlation
with colour (r: 0.48, p<0.001, all section; Fig. S1) or calcium carbonate (r: 0.36, p<0.001,
between C35 and C44; Fig. S1). Therefore, the Santiurde relationship suggests that the
MS signal is more likely controlled by ferromagnetic minerals, such as magnetite (Fig.
S2).
**4.1.3. Time series analysis**
Prior to spectral analysis, the colour data series was regularly interpolated (spacing of
0.06 m) and the 3$^{rd}$ order polynomial trend was subtracted. The 2π-MTM power spectrum
of the colour data series shows peaks at four period bands: 30-42 cm (peaking at 37 cm),
1 m, 1.67 m and 5-10 m (Fig. 4). The short period band shows significant peaks above
99% CL. In the intermediate period band, the 1 m peak exceeds 95% CL and the 1.67 m
peak reaches 90% CL. The long period band, which peaks at 6.6 m, is above 99% CL.
The short period band matches the average thickness of couplets and the longest
intermediate band the average thickness of bundles. The EHA and wavelet spectra also
highlight the four main period bands, although the 1-m-periodicity is relatively less
relevant. The period bands are not continuous and there are several intervals where the
signal loses power, such as the 11-16 m and 24-36 m intervals of the short period band.
Spectral analysis carried out on MS data corroborate the prevalence of the
abovementioned four period bands, although the intermediate bands do not reach high
confidence levels (Fig. S3).
Using the average values of the period bands identified by spectral analysis, the short and
long intermediate period components were separately extracted from the colour data
series through Gaussian bandpass filtering (Fig. 5). The number of oscillations in the short
period filter matches the number of couplets defined in the outcrop and in the colour
curve. Similarly, the oscillations in the intermediate period filter match the number and
thickness of bundles.





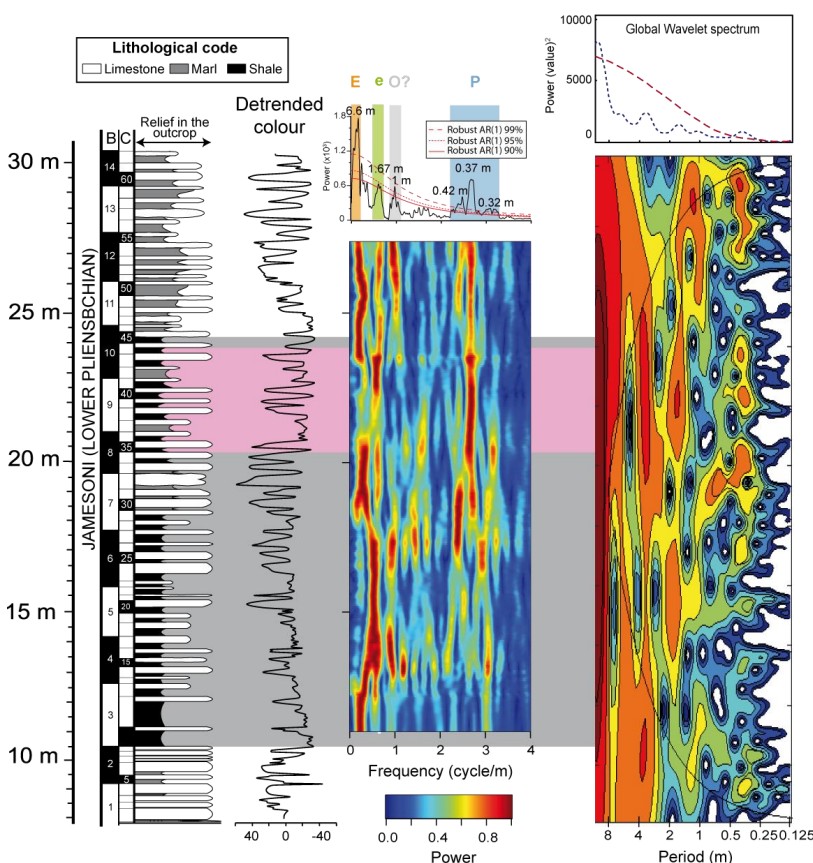

Figure 4. Stratigraphic log and chronostratigraphy of the studied section, showing the detrended colour curve. Bundles
(B) and couplets (C) identified in the sedimentary alternation are numbered in ascending stratigraphic order. The grey
background shows the extent of the Uptonia jamesoni Black Shale 1, and the pink interval in its upper part shows the
interval studied herein in detail. The 2π-MTM, EHA and Wavelet spectra of the colour data series show the occurrence
of four main period bands: 30-42 cm (in blue in the 2π-MTM spectrum), interpreted as precession (P) couplets; 1 m
(grey), possibly related to obliquity (O?) cycles; 1.67 m (green), representing short eccentricity (e) bundles; and 5-10
m (peak at 6.7 m; orange), which corresponds to long eccentricity (E) bundles.

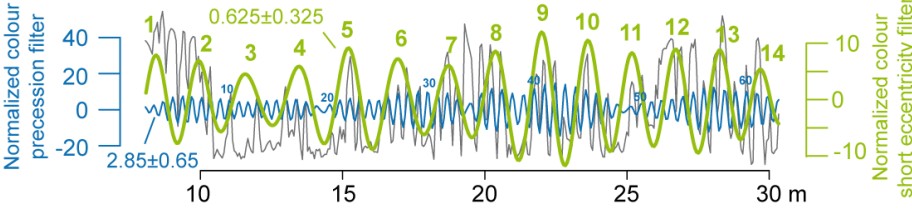

Figure 5. Colour filter outputs of short (in blue) and intermediate (green) period bands, which are related to precession
couplets and short eccentricity bundles respectively.
**4.2. Detailed analysis of Bundle 9 (C35-C44 interval)**
**4.2.1. L/M ratio and calcium carbonate content**



The limestone/marlstone (L/M) thickness ratio of couplets varies between 0.33 (C42) and
1.36 (C39), with an average value of 0.90 (Fig. 6A, Table S2). The highest L/M values
are found in the couplets at the central part of bundle B9, while the lowest values
correspond to couplets C41 and C42, at the boundary between bundles B9 and B10.
The $CaCO_3$ content ranges from 24.63 to 88.97%, averaging out at 49.78% (Fig. 6B;
Table S3). In general, %$CaCO_3$ fluctuates in line with lithology, limestones and marly
limestones (average: 66.36%) being richer than marls and shales (average: 34.86%).
Marls and shales differ by 10-15% in their $CaCO_3$ content, whereas limestone beds at the
central part of bundle B9 show 20-40% more $CaCO_3$ than counterpart marly limestones
at bundle boundaries.

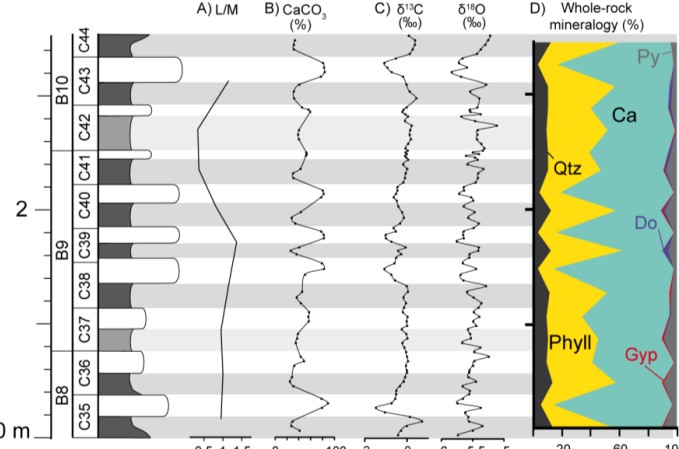

Figure 6. Lithological log of the Santiurde interval studied in detail (dark grey: shale; pale grey: marl; white: limestone
and marly limestone), showing (A) the limestone–marl (L/M) thickness ratio of couplets, (B) %$CaCO_3$ content, (C)
$\delta^{13}C_{carb}$ and $\delta^{18}O_{carb}$ curves and (D) whole-rock mineralogy. Numbered couplets and bundles are labelled C and B,
respectively. The 0 m level corresponds to 20.34 m of the general section.

## 4.2.2. Carbon and oxygen isotopes

$\delta^{13}C_{carb}$ values range from -1.5 (C35L) to 0.70‰ (C35M) and average out at -0.25‰ (Fig.
6C). The $\delta^{13}C_{carb}$ curve shows lower values in limy beds and higher values in shales and
marls. The amplitude of the fluctuations is significantly greater in the central couplets of
bundle B9. $\delta^{18}O$ values range from -5.84 (C43L) to -5.25‰ (C36L) and average out at -
5.52‰, the $\delta^{18}O$ curve being rather spiky. $\delta^{13}C_{carb}$ and $\delta^{18}O_{carb}$ data show intermediate
positive correlation (r: 0.53; $p<0.005$; Fig. S4A; Table S3).

## 4.2.3. General mineralogy

XRD results (Fig. 6D; Table S2) show that calcite is the most abundant mineral in limy
beds and in some of the marl/shales (28 to 84%, average: 54%). Clay minerals constitute
the second most abundant phase (9 to 50%, average: 32%), followed by quartz (3 to 13%,
average: 9%) and other minor components (pyrite, gypsum, and dolomite).





The mineralogical content fluctuates in line with lithology, as it shows maximum values
of clays and quartz, and minima of calcite, in marls/shales. Moreover, the amplitude of
the detrital/carbonate mineralogical oscillations increases in the central couplets of bundle
B9. Pyrite, despite being a minor component (0.5 to 9%, average: 4%), also oscillates
with lithology, presenting maximum values in marls/shales, but does not match the
amplitude variation associated with the bundle arrangement.
**4.2.4. Organic matter geochemistry**
The content in organic carbon varies between 0.26 (C39L) and 4.03% (C41M) (average
of 1.91%), maximum values being found at black shales. Organic nitrogen also covaries
with lithology, with values ranging from 0.02 (C39L) to 0.09 % (C36M) (average of
0.06%). Both elements show high amplitude oscillations at the central part of bundle B9
and subdued oscillations at bundle boundaries. The relationship between both organic
components was calculated by the C/N ratio (Fig. 7; Table S2)
$\delta^{13}C_{org}$ values vary between -29.6 (C40M) and -27.2‰ (C40L), and average out at -
28.6‰. $\delta^{15}N_{org}$ ranges from 1.1 (C38L) to 3.2‰ (C40M), with an average value of 2.5‰
(Fig. 7). Both data series alternate in line with lithology, but with opposite trends. The
$\delta^{13}C_{org}$ fluctuations at the central couplets of bundle B9 show the greatest amplitude.

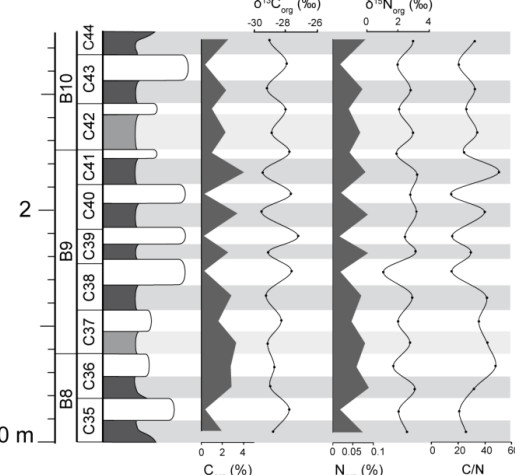

Figure 7. Lithological log of the Santiurde interval studied in detail, showing fluctuations in the percentage of organic
C and N, C/N ratio, $\delta^{13}C_{org}$ and $\delta^{15}N_{org}$.
**4.2.5. Elemental geochemistry**
The average abundance of major and trace elements is shown in Fig 8 (Table S4). $SiO_2$,
$Al_2O_3$ and CaO constitute 48% of limestones and 63% of marls/shales. Average values
of most major and trace elements are higher in marls and shales than in limy beds, the
exceptions being CaO, MnO, Ba and Sr. The correlation matrix shows that the abundance
of MnO does not correlate with any major and trace elements, but all the other major
elements present strong negative correlation (>-0.88) with CaO (Table 1) and high
positive correlation with most redox sensitive trace elements (Co, Cu, Ni, V and Zn), the





only exception being Zn, which shows intermediate positive correlations. Sr and Ba
display intermediate positive correlation with each other.

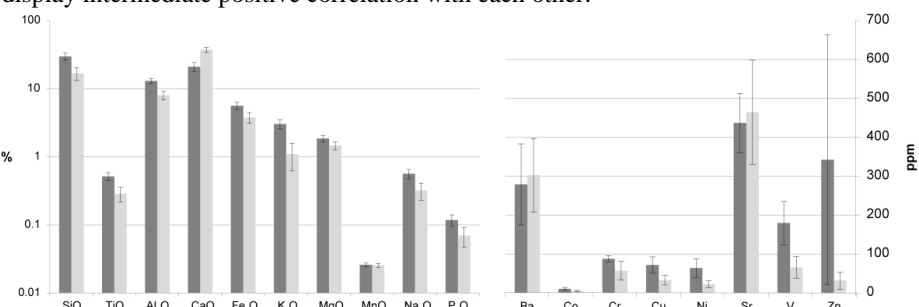

Figure 8. The average abundance of major and trace elements of limestones (pale grey) and marl/shales (dark grey).

| | SiO2 | TiO2 | Al2O3 | CaO | Fe2O3 | K2O | MgO | MnO | Na2O | P2O5 | Ba | Co | Cr | Cu | Ni | Sr | V | Zn |
|---|---|---|---|---|---|---|---|---|---|---|---|---|---|---|---|---|---|---|
| SiO2 | | 9.9E-15 | 1.2E-09 | 9.2E-16 | 5.8E-07 | 1.3E-11 | 2.9E-07 | 0.97 | 2.3E-06 | 1.1E-06 | 6.1E-01 | 2.6E-07 | 2.0E-10 | 2.1E-06 | 4.3E-05 | 5.8E-01 | 2.8E-03 | 2.3E-02 |
| TiO2 | **0.99** | | 4.0E-10 | 4.2E-15 | 4.3E-08 | 9.4E-14 | 7.1E-08 | 0.80 | 4.8E-06 | 7.0E-08 | 5.6E-01 | 7.5E-07 | 2.2E-09 | 8.4E-06 | 7.9E-05 | 6.5E-01 | 8.9E-04 | 2.0E-02 |
| Al2O3 | **0.94** | **0.95** | | 1.4E-12 | 6.2E-09 | 1.3E-12 | 2.8E-06 | 1.00 | 5.4E-07 | 1.7E-05 | 9.1E-01 | 3.2E-04 | 4.7E-09 | 5.1E-06 | 1.1E-03 | 3.5E-01 | 1.9E-04 | 1.4E-02 |
| CaO | **-0.99** | **-0.99** | **-0.98** | | 1.4E-08 | 5.3E-15 | 1.8E-07 | 0.96 | 5.7E-07 | 9.5E-07 | 6.6E-01 | 4.7E-06 | 2.4E-11 | 9.4E-06 | 6.2E-05 | 5.5E-01 | 4.4E-04 | 1.7E-02 |
| Fe2O3 | **0.88** | **0.91** | **0.93** | **-0.93** | | 6.2E-09 | 1.8E-06 | 0.57 | 7.4E-07 | 8.1E-06 | 6.9E-01 | 1.1E-03 | 1.1E-06 | 3.3E-04 | 6.2E-04 | 8.2E-01 | 4.5E-06 | 1.0E-02 |
| K2O | **0.97** | **0.98** | **0.98** | **-0.99** | **0.93** | | 1.8E-06 | 0.97 | 1.1E-06 | 6.1E-07 | 7.1E-01 | 4.5E-09 | 3.6E-05 | 1.3E-04 | 1.3E-04 | 5.8E-01 | 1.3E-04 | 2.0E-02 |
| MgO | **0.89** | **0.91** | **0.86** | **-0.90** | **0.86** | **0.86** | | 0.10 | 0.00 | 0.00 | 0.52 | 0.00 | 0.00 | 0.00 | 0.00 | 0.67 | 0.00 | 0.14 |
| MnO | -0.01 | 0.06 | 0.00 | -0.01 | 0.14 | 0.01 | 0.39 | | 0.85 | 0.62 | 0.68 | 0.77 | 0.75 | 0.60 | 0.75 | 0.82 | 0.46 | 0.22 |
| Na2O | **0.86** | **0.85** | **0.88** | **-0.88** | **0.88** | **0.87** | **0.79** | 0.05 | | 0.00 | 0.34 | 0.01 | 0.00 | 0.00 | 0.00 | 0.90 | 0.00 | 0.02 |
| P2O5 | **0.87** | **0.91** | **0.82** | **-0.87** | **0.84** | **0.88** | **0.80** | 0.12 | **0.72** | | 0.53 | 0.00 | 0.00 | 0.00 | 0.58 | 0.00 | 0.01 |
| Ba | -0.13 | -0.14 | 0.03 | 0.11 | -0.10 | -0.09 | -0.16 | -0.10 | -0.23 | -0.16 | | 0.43 | 0.55 | 0.24 | 0.26 | 0.00 | 0.29 | 0.91 |
| Co | **0.89** | **0.88** | **0.74** | **-0.85** | **0.69** | **0.82** | **0.75** | -0.07 | ***0.61*** | **0.85** | -0.19 | | 7E-05 | 1E-09 | 5E-07 | 7E-01 | 8E-02 | 2E-02 |
| Cr | **0.96** | **0.94** | **0.93** | **-0.97** | **0.87** | **0.93** | **0.91** | 0.08 | **0.88** | **0.80** | -0.15 | **0.79** | | 0.00 | 0.00 | 0.66 | 0.00 | 0.10 |
| Cu | **0.86** | **0.84** | **0.72** | **-0.83** | **0.74** | **0.80** | **0.71** | -0.13 | **0.69** | **0.84** | -0.28 | **0.94** | **0.77** | | 1E-09 | 1E+00 | 2E-02 | 4E-02 |
| Ni | **0.80** | **0.78** | **0.69** | **-0.79** | **0.71** | **0.77** | **0.67** | -0.08 | **0.65** | **0.78** | -0.27 | **0.88** | **0.75** | **0.95** | | 0.9 | 0.0 | 0.0 |
| Sr | 0.14 | 0.11 | 0.23 | -0.15 | 0.06 | 0.10 | 0.11 | 0.05 | -0.03 | 0.14 | **0.65** | 0.08 | 0.11 | -0.01 | -0.04 | | 0.49 | 0.98 |
| V | **0.65** | **0.70** | **0.76** | **-0.73** | **0.85** | **0.77** | **0.65** | 0.18 | **0.82** | **0.65** | -0.26 | 0.42 | **0.71** | ***0.52*** | ***0.62*** | -0.17 | | 0.02 |
| Zn | ***0.52*** | ***0.53*** | ***0.55*** | ***-0.54*** | ***0.57*** | ***0.56*** | 0.35 | -0.29 | ***0.52*** | ***0.59*** | -0.03 | ***0.53*** | 0.39 | ***0.62*** | **0.65** | 0.01 | ***0.52*** | |


Table 1. Pearson correlation coefficient (r) of major and trace element concentrations in the lower left part of the matrix.
The p-value for each coefficient is located in the upper right part of the matrix. Highest (r > ±0.65) correlations are
marked in bold and intermediate correlations (r ≥ ±0.50-0.64) in bold and italics.
In order to compare the abundance of some elements with the reference average shale
composition (Li and Schoonmaker, 2003), enrichment factors ($X_{EF}$; Tribovillard et al.,
2006) were calculated as follows: $X_{EF} = (X/Al)_{sample}/(X/Al)_{average\ shale}$. Al and K are
commonly thought to be related to the clay fraction, whereas Si and Ti are often
associated with the coarser fraction of quartz and heavy minerals (Calvert and Pedersen,
2007). Enrichment in Ti has also been related to stronger aeolian input (Rachold and
Brumsack, 2001). In Santiurde $K_{EF}$, $Ti_{EF}$ and $Si_{EF}$ covary with lithology, showing
maximum values in marls/shales and increasing the amplitude of the oscillations in the
middle part of bundle B9 (Fig. 9).
Marine palaeoproductivity is commonly associated with algal growth, which varies with
the availability of macro-nutrients, such as P and N (Calvert and Pedersen, 2007). $P_{EF}$
values from Santiurde show that these deposits are depleted in P (Li and Schoonmaker,
2003). However, $P_{EF}$ shows higher values in marls/shales than in limy beds in almost all
couplets (except in C35L and C43L; Fig. 9). Authigenic Ba in marine sediments is
commonly associated to barite and its abundance is generally determined by organic C
export from surface waters into deep marine environments (Tribovillard et al., 2006). In



order to minimize the influence of detrital barium in palaeoenvironmental analyses, $Ba_{EF}$
and the $Ba_{excess}$ index are widely used (Dymond et al., 1992). $Ba_{EF}$ shows that the studied
succession is significantly depleted in Ba in comparison with average shales (Fig. 9, Li
and Schoonmaker, 2003). Both $Ba_{EF}$ and $Ba_{excess}$ reveal increased accumulation of Ba
when OM-poor limestones were deposited, just the opposite of $P_{EF}$.

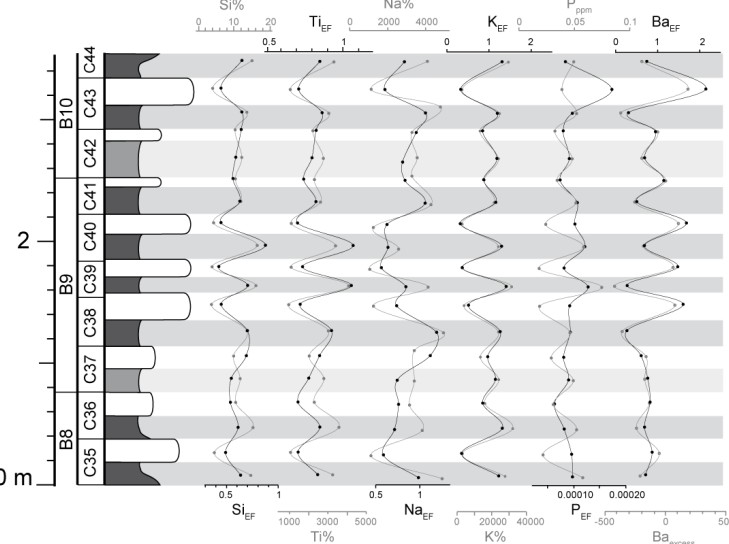

Figure 9. Lithological log of the Santiurde interval studied in detail, showing fluctuations in the percentage of elements
related to detrital input (Si, Ti, Na and K), palaeoproductivity (P and Ba) and their enrichment factors (EF). The $Ba_{excess}$
ratio is also presented.
$Mn_{EF}$ is commonly linked to authigenic Mn phases, such as authigenic oxi-hydroxides.
In Santiurde $Mn_{EF}$ shows an oscillatory pattern in line with lithology, with maximum
values at limestones (Fig. 10). As no evidence of Pliensbachian hydrothermal or volcanic
activity has been reported to date in the area, the higher $Mn_{EF}$ in limestones could suggest
increased terrestrial input, more oxygenated deep waters or increased remineralization of
organic matter (Bayon et al., 2004; Tribovillard et al., 2006; Calvert and Pedersen, 2007).
Both V and Zn commonly show a strong association with OM content (Calvert and
Pedersen, 2007; Algeo and Liu, 2020). The type of organic matter affects the distribution
of both elements, as V is taken up by tetrapyrrhole complexes derived from chlorophyll
decay, whereas Zn is known to be incorporated into humic and fulvic acids (Lewan, 1984,
Aristilde et al., 2012). Enrichment factors of both elements show oscillatory patterns in
line with lithology, with maximum values at shales/marls and a significant enrichment in
V (Fig. 10). On the other hand, Co, Cu and Ni are known to be related with sulphide
fractions (Tribovillard et al., 2006; Algeo and Liu, 2020), as these elements are usually
incorporated as minor constituents in diagenetic pyrite (Berner et al., 2013). With the
exception of $Cu_{EF}$, the enrichment factors of these elements also fluctuate with the
lithological alternation, showing maximum values in shales/marls (Fig. 10).
Several bielemental ratios associated with redox conditions during sedimentation were
also calculated. According to absolute values of the V/Cr bielemental ratio, most marls

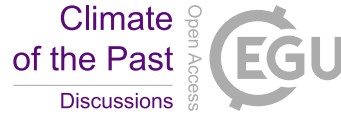



and shales deposited under dysoxic conditions, whereas limestones and marly limestones
were accumulated in oxic conditions (Fig. 10; Jones and Manning, 1994). Ni/Co values
from marls and shales support dysoxic or even suboxic/anoxic conditions (Fig. 9, Jones
and Manning, 1994), but suggest that limestones and marly limestones also accumulated
in nearly dysoxic conditions. The discrepancy between V/Cr and Ni/Co results confirms
the limitation of absolute bielemental ratios to discriminate absolute redox conditions
(Algeo and Liu, 2020). In Santiurde all lithologies are enriched in V, Zn, Ni and Cu when
compared with average shales (Li and Schoonmaker, 2003). The concentration of these
redox-sensitive trace elements is generally higher than in crustal rocks when sediments
accumulate under oxygen depleted conditions (Brumsack, 1986; Arthur et al., 1990).
Consequently, it is assumed that deep seawater oxygen concentration was fluctuating, but
the general background conditions of the environment were depleted in oxygen.

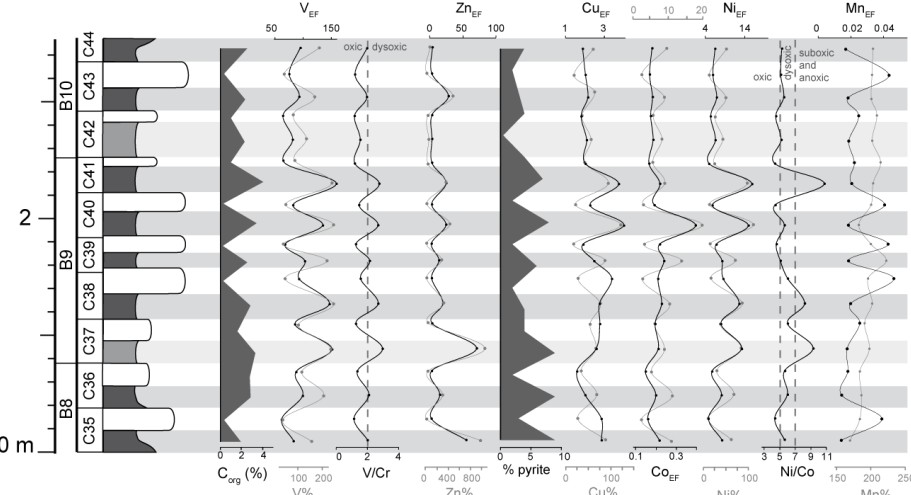

Figure 10. Lithological log of the Santiurde interval studied in detail, showing fluctuations in the percentage of redox
sensitive elements, their EFs and several bielemental ratios, along with the organic carbon and pyrite content.
**4.2.6. Factor analysis**
A statistical factor analysis was conducted in order to identify key groups of variables
with similar trends in the mineralogical and geochemical databases. As the number of
variables introduced in the analysis has to be lower than the number of cases (samples),
a total of 18 variables of the total dataset were selected. To this end, elements with very
strong mutual correlation coefficients (for example, Mg and Fe with Al) were ignored,
because they would yield redundant data and increase the size of the dataset. Main redox
sensitive elements, in whose Santiurde is enriched, have been included because their
palaeoenvironmental significance. Variables with no quantifiable concentrations
throughout the studied section (e.g., gypsum and dolomite content) were also excluded.
Thus, the analysed dataset consists of 18 variables (see Table S5 and Fig. 11) from 19
cases (beds).



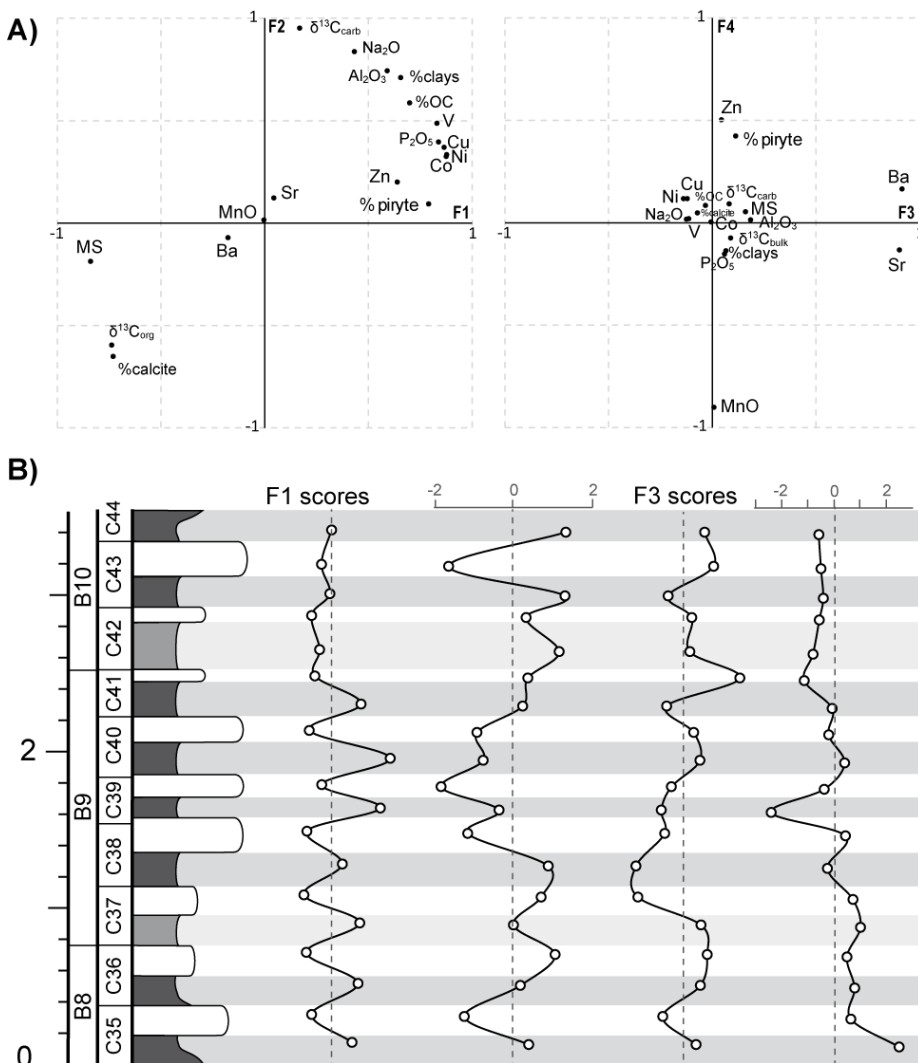

Figure 11. A) Projection of different elements in the Factor 1 versus Factor 2 cross-plot (ca. 70% of the total variance) and in the Factor 3 versus Factor 4 cross-plot (ca. 18% of the total variance). B) Stratigraphic distribution of factorial scores of the four extracted factors (virtual variables).

The optimal factor analysis (varimax rotation) extracted four factors (F1 to F4) or virtual variables that have eigenvalues greater than one. These factors explain 87.97% of the cumulative variance of the analyzed data matrix (Fig. 11 and Table S5). Factors 1 and 2 explain the highest percentage of the dataset, 44.54% and 25.78% respectively. Both factors explain the variance of variables linked to the lithological alternation and the arrangement of couplets in bundles (Fig. 11). F1 shows higher loadings for variables linked to oxygenation state (trace elements, pyrite, $C_{org}$ vs $\delta^{13}C_{org}$, MS) and palaeoproductivity ($P_2O_3$). Conversely, F2 has higher loadings in variables ($Na_2O$, $Al_2O_3$, clay% vs calcite) linked to the dilution of calcite with terrigenous material; $\delta^{13}C_{carb}$ also



shows a very high positive loading with F2. Factors 3 and 4 explain a significantly lower
variance of the total dataset, 9.92 and 7.73% respectively. F3 shows very high positive
loadings for Ba and Sr, whereas F4 shows very high negative loadings for MnO and
intermediate positive loading for Zn and pyrite. The scores of factors 3 and 4 do not align
with the lithological arrangement in couplets and bundles, which suggests that they were
not controlled by orbitally influenced environmental conditions.
**5. Discussion**
**5.1. Origin of the inorganic sedimentary fluctuations**
**5.1.1. Santiurde rhytmites: primary or diagenetic?**
Previous studies have shown that the formation of calcareous rhytmites can be caused by
both primary and diagenetic processes. In some cases, rhytmites have been considered to
be primary, being related to secular variations in the environmental conditions that
controlled sedimentation (e.g., Arthur and Dean, 1991; Hinnov and Park, 1999; Dinarés-
Turell et al., 2018; Martinez-Braceras et al., 2023). In other cases, postdepositional
dissolution/cementation processes have been considered the most important (e.g.,
Hallam, 1986; Reuning et al., 2002; Westphal, 2006; Nohl et al. 2021). When differential
diagenesis affects the primary composition of sediments, part of the carbonate dissolves
from marly beds and migrates to limy beds, precipitating as cements (Westphal, 2006).
The Santiurde deposits show some signs of diagenetic overprinting, such as the
occurrence of some secondary cements, calcite overgrowths, early framboidal pyrite and
the growth of pyrite in tests (Fig. 3). In fact, the aragonitic and high-Mg calcite
components of limestones, including their micritic matrix, suffered significant re-
crystallization. However, none of the limestone beds displays a nodular geometry, which
is common in successions affected by intense postdepositional dissolution/cementation
processes (Hallam, 1986; Einsele and Ricken, 1991). Quite the opposite, the
characteristics of the beds are continuous for more than 1 km between the Santiurde
motorway and railway sections. Furthermore, petrographic and SEM observations
suggest that fluid migration from marly to limy beds was overall limited. Thus, skeletal
components of marls/shales (Fig. 2 and 3) do not present features of increased compaction
(Munnecke et al., 2001; Westphal, 2006). This was probably related to an original higher
clay content in marls/shales, which hampered fluid migration between beds and avoided
intense dissolution and recrystallization. In addition, clay minerals show primary textures
(such as deformed, broken plates or isolated flakes wrapping other detrital grains), but do
not show any evidence of intense diagenetic recrystallization. In general, the diagenetic
characteristics observed in the Santiurde rhytmites are typical of processes related to
organic matter decay during burial (Rosales et al., 2001).
Interestingly, the lithological arrangement in couplets and bundles observed in the
outcrop, combined with the spectral analysis of colour and MS data series, highlight the
presence of sedimentary cycles with three main periodicities in the succession
(6.6:1.67:0.36). This ratio is comparable to the 405:100:20 ratio produced by the



superposition of long eccentricity, short eccentricity and precession cycles (Berger and
Loutre, 1994).
The abovementioned characteristics strongly suggest that the formation of the Santiurde
rhythmites was primary and responded to orbitally driven climate change episodes. An
orbital control on sedimentation had previously been deduced in other Pliensbachian
successions from nearby areas, such as the Asturian and Iberian basins (Bádenas et al.,
2012; Val et al., 2017; Sequero et al., 2017).

### 5.1.1. Preservation of the geochemical signal

Although the formation of the Santiurde rhytmites was a result of orbitally paced
environmental variations, some primary sedimentary characteristics (such as chemical
and mineralogical composition, fossil assemblage, or porosity) could have responded in
different ways to diagenesis. Consequently, the geochemical data of the seven limestone-
marl couplets (C35-C44) studied in detail must be analyzed carefully in order to interpret
which environmental variations controlled sedimentation.
Whole-rock inorganic isotopic analyses from diagenetically "closed" systems, such as
hemipelagic carbonates, have been used successfully for the climatic reconstruction of
ancient sedimentary environments (e.g., Jenkyns and Clayton, 1986; Marshall, 1992;
Silva et al., 2011; Martínez-Braceras et al., 2017; Deconinck et al., 2020). However,
$\delta^{13}C_{carb}$ and $\delta^{18}O_{carb}$ values tend to get depleted during burial, causing a significant
positive correlation between each other when strong deep burial or meteoric diagenesis
affects the succession (Banner and Hanson, 1990; Marshall, 1992; Swart, 2015). In
Santiurde both isotopic records show depleted values in comparison to Early Jurassic
marine isotopic standard curves (Grossman and Joachimski, 2020; Cramer and Jarvis,
2020). Both $\delta^{18}O_{carb}$ and $\delta^{13}C_{carb}$ records show a positive but not very high correlation
(Fig. S4A; r: 0.53, p<0.005), following a common burial trend (Banner and Hanson,
1990). This suggests that, although primary isotopic trends may have been preserved,
absolute values are probably distorted. Accordingly, $\delta^{18}O_{carb}$ values from Santiurde are
significantly depleted (Grossman and Joachimski, 2020) and display a spiky curve (Fig.
6). This may reflect the impact of the percolation of diagenetic fluids in post-depositional
processes at low fluid/rock ratios (Banner and Hanson, 1990). Consequently, $\delta^{18}O_{carb}$
values were only used to assess the degree of diagenetic overprinting of other
geochemical proxies.
Rosales et al. (2001) analyzed the utility of stable isotopes from Lower-Middle Jurassic
bulk hemipelagic carbonates and fossils (belemnites and brachiopods) from the BCB as
palaeoceanographic proxies. They concluded that whole rock stable isotope records are
not suitable for accurate palaeoceanographic reconstructions because their high OM
content contributed to the alteration of their primary signal. In fact, organic matter
degradation and sulphate reduction in deep sea sediments is known to produce $CO_2$
enriched in $^{12}C$ and generate early cements with low $\delta^{13}C_{carb}$ (Dickson et al., 2008; Swart,
2015). Accordingly, the generally depleted $\delta^{13}C_{carb}$ values in Santiurde could be a
consequence of the addition of early cements precipitated in equilibrium with isotopically

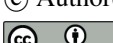



light pore waters affected by OM decay. This process, however, cannot explain the
$\delta^{13}C_{carb}$ fluctuations observed along the lithological alternation, because the influence of
$\delta^{13}C$-depleted fluids is generally thought to be more pronounced when carbonate content
in the sediment is low and the total organic carbon is comparatively high (Ullman et al.,
2022). Contrarily, in Santiurde maximum $\delta^{13}C_{carb}$ values are recorded in marls/shales and
the crossplot of $\delta^{13}C_{carb}$ versus $CaCO_3$ values shows a high negative correlation (r: -0.75,
p<0.005; Fig. S4B). It can therefore be assumed that the high clay content and low
porosity in marls/shales probably hampered a more intense cementation during early
diagenesis (Arthur and Dean, 1991).
In line with the above argumentation, $\delta^{13}C_{carb}$ records of hemipelagic carbonates are
commonly used in palaeoclimatic studies because they are not strongly affected by the
bicarbonate composition and temperature of interstitial waters (Marshall, 1992;
Mackensen and Schmiedl, 2019). However, dissolution of aragonite and high-Mg calcite
components, which are generally more abundant in shallow marine areas, and
precipitation of more stable low-Mg calcite phases are important post-depositional
process causing carbon isotope fractionation (Reuning et al., 2002). Aragonite is
generally characterized by more positive $\delta^{13}C$ values than high- or low-Mg carbonates
(Swart, 2015). Therefore, a fluctuating rate of aragonitic input could produce covarying
$\delta^{13}C_{carb}$ and %$CaCO_3$ records (Reuning et al., 2002), like that found in Santiurde.
However, given that minimum $\delta^{13}C_{carb}$ values are found at %$CaCO_3$ maxima in Santiurde,
it can be concluded that the carbonate distribution does not record variations in the supply
of platform-derived fine-grained aragonitic and high-Mg calcite.
Whole rock $\delta^{13}C$ and $\delta^{18}O$ average values similar to those obtained in Santiurde were also
found in the coeval Rodiles hemipelagic section from the Asturian basin (Deconinck et
al., 2020), that isotopic trend being considered to reveal primary environmental changes.
In fact, $\delta^{18}O_{carb}$ values from Santiurde are within the average range of those obtained from
Pliensbachian belemnites from the Asturian basin (Gómez et al., 2016; Armendáriz et al.,
2012), which were used for palaeoceanographic reconstructions. Taking everything into
account, it can be concluded that the $\delta^{13}C_{carb}$ record from Santiurde may reflect the
original isotopic composition of seawater, but it cannot be excluded that the fluctuations
respond to original variations in the isotopic signal of pore waters. However, the
elemental geochemical evidence further suggests that, in addition to the original
composition and porosity of the different layers, the Santiurde rhythmites also records
variations in the supply of terrigenous components. Thus, diagenetically inert trace
elements, such as $Ti_{EF}$, also show variations in line with the lithological alternation (Nohl
et al., 2021).
Other elements, such as Sr, Fe and Mn, are sensitive to burial and may be used to assess
the degree of diagenetic overprinting in carbonates in combination with $\delta^{18}O_{carb}$ values
(Marshall, 1992; Rosales et al., 2001; Zhao and Zheng, 2014). In general, during
diagenesis, marine carbonates tend to become depleted in Sr and $\delta^{18}O$, but enriched in Fe
and Mn (Banner and Hanson, 1990). There is no correlation between the abundance of
these three elements in Santiurde (Fig. S5; Sr-Mn r: 0.03, p: 0.9; Sr-Fe r: 0.06, p: 0.82;



Mn-Fe r: 0.14, p: 0.58). Moreover, $\delta^{18}O_{carb}$ values do no display any correlation with Sr
and Mn and show positive correlation with Fe, just the opposite of what should be
expected from postdepositional distortion. Similarly, if compared with the average shale
composition (Li and Schoonmaker, 2003), both limestones and marls from Santiurde are
significantly enriched in Sr (402.5 ppm), slightly enriched in Fe (32750 ppm), and
depleted in Mn (199 ppm). Taking everything into account a strong diagenetic
overprinting can be ruled out.
In conclusion, burial diagenesis produced depleted inorganic stable isotope values, but
there are no signs of strong differential diagenesis or postdepositional redistribution of
geochemical components in the Santiurde section. The $\delta^{13}C_{carb}$ signal was affected by
early diagenetic processes related to OM decay in limestones, but not to the extent of
obscuring the original fluctuating trend.

### 587  5.2. Organic matter: fluctuating composition and preservation

Detailed multiproxy analysis carried out throughout 7 limestone-marl couplets from the
oldest BS cast light on the origin of OM and the sedimentary factors that controlled its
distribution. Rosales et al. (2006) showed that BS intervals accumulated during second
order sea level rises, which originated the flooding of large continental areas and the
creation of a moderately isolated epicontinental sea, in which water circulation was
relatively restricted. More specifically, sluggish circulation at the depocentres of the
irregular floor of the BCB contributed to increasing density stratification of the water-
column and caused a sea floor depleted in oxygen (Wignall, 1991; Quesada et al., 2005),
which prevented oxidation of the high organic matter content of the section.

### 597  5.2.1. Composition of OM

Previous studies demonstrated that the greatest part of the organic matter found in the
BCB Pliensbachian black shales had a marine origin, being dominated by amorphous and
structured liptinitic organic matter (Suárez-Ruiz and Prado, 1987; Quesada et al., 1997,
2005; Permanyer et al., 2013). The study of saturated biomarkers corroborated a dominant
pattern of mature extracts derived from marine algal components. Additionally, SEM
analysis carried out in the present study provided evidence of the occurrence of biofilms
with sporadic occurrences of vitrinite (Fig. 3E and F).
The average organic C/N ratio of 30.45 obtained in Santiurde (Fig. 7) is significantly
higher that of modern marine organic matter, which usually displays values between 5
and 18 (Meyers, 2006). However, C/N ratios observed in current reservoirs cannot be
directly extrapolated to ancient rocks, especially to those deposited under high
productivity conditions (Nijenhuis and Lange, 2000; Meyers et al., 2006; Schneider-Mor
et al., 2012). Meyers et al. (2006) observed that organic components from Albian to
Santonian black shales from Demerara Rise were mainly marine in origin, but their C/N
ratio varied between 20 and 45, which is commonly assigned to terrestrial plants. Those
high C/N values were related to a more rapid recycling of N than C during OM
decomposition. Under oxic to anoxic conditions, modern marine organic matter is



commonly degraded via denitrification, decomposing principally nitrogen-rich
aminoacids and reducing the total organic N of sediments (Altabet et al., 1995; Van Mooy
et al., 2002). Thus, high C/N values of some Mediterranean sapropels and Cretaceous
black shales have been related to the drawdown of dissolved oxygen in the water column
under conditions of high export productivity (Nijenhuis and Lange, 2000; Schneider-Mor
et al., 2012). Similar processes might have controlled OM degradation in Santiurde,
producing the abovementioned high C/N ratio. In this regard, considering that the C/N
ratio of typical marine OM is closer to ~6, at least ~23% of the original N must have been
removed from the Santiurde deposits due to denitrification. This percentage is higher than
that calculated by experimentation (~9%) in recent sediments (Van Mooy et al., 2002),
but significantly lower than the 70% deduced from Cretaceous indurate successions
(Schneider-Mor et al., 2012). This suggests that other processes related to OM
degradation, as well as the duration of the process, determine the loss of N due to
differential degradation.
The $\delta^{13}C_{org}$ signal from Santiurde is also relatively depleted if compared to modern
marine OM, being closer to values of terrestrial plants (Schneider-Mor et al., 2012).
However, similarly depleted $\delta^{13}C_{org}$ values of marine OM have also been found in other
indurate successions (Nijenhuis and Lange, 2000; Schneider-Mor et al., 2012). This
general depletion of $\delta^{13}C_{org}$ compared to average algal tissue is associated with selective
decomposition of carbohydrates and proteins enriched in $^{13}C_{org}$, which are more easily
decomposed, and the fortification of the lipid fraction enriched in $^{12}C_{org}$ (Jenkyns and
Clayton, 1986). A similar fractionation process was invoked in other sections, such as the
Cretaceous oil shales from Israel (Schneider-Mor et al., 2012) and the Mediterranean
Pliocene sapropels (Nijenhuis and Lange, 2000).
In conclusion, poorly oxygenated background conditions of bottom waters triggered
denitrification of marine OM in Santiurde, promoting a selective decomposition of
nitrogen-rich aminoacids and the fraction enriched in $^{13}C_{org}$. This process may have been
stronger during the deposition of OM-rich shales.

### 5.2.2. Fluctuations in OM content and characteristics

In Santiurde, the OM content fluctuates in line with lithology, suggesting that the
environmental factors that controlled its accumulation and/or preservation varied
cyclically (Fig. 7). The fluctuations in OM content could be the result of variations in
either the flux of organic matter to the sea floor (i.e., fluctuations in productivity), or the
rate of dilution by terrestrial or carbonate sedimentary inputs, or the rate of organic-matter
remineralization (i.e., fluctuations in preservation) due to changing seawater oxygen
concentrations (Tyson, 2005; Swart et al., 2019).
Many factors affect sedimentary $\delta^{13}C_{org}$ values of marine sediments, such as biological
sources, recycling of organic matter, and marine productivity (e.g., Nijenhuis and Lange,
2000; Tyson, 2005; Meyers et al., 2006; Luo et al., 2014). Changes in marine productivity
can be ruled out for the Santiurde $\delta^{13}C_{org}$ fluctuations. Indeed, increased OM production
generally results in greater sequestration of $^{12}C$, which would originate higher $\delta^{13}C_{org}$





values when OM content increased (Meyers et al., 2006), just the opposite of the Santiurde
trend (Fig. 7). This is also confirmed by $\delta^{15}N_{org}$ values, which can also be subject to
fractionation due to variations in productivity. N is assimilated by organisms in order to
produce biomass, preserving the $\delta^{15}N_{org}$ value of its source. Marine $\delta^{15}N_{org}$ values are
influenced by changes in ocean circulation, biological pump, large scale N cycling, and
redox conditions (Robinson et al., 2012). Nitrogen isotopes have been used as a powerful
tool in the analysis of petroleum systems in order to evaluate unconventional reservoirs,
deduce palaeoenvironmental conditions, and assess organic matter sources (Quan and
Adeboye, 2021). However, $\delta^{15}N_{org}$ values may also be subject to distortions during
sedimentation, burial diagenesis, catagenesis and hydrocarbon migration (Robinson et al.,
2012; Quan and Adeboye, 2021). Average $\delta^{15}N_{org}$ values from Santiurde (Fig. 7) are close
to the current ocean isotopic ratio (~5‰; Robinson et al., 2012) and vary within the range
observed in other organic-rich sediments and rocks (principally shales and marlstones;
Holloway and Dahlgren, 2002). Increased N fixation rates have been observed in modern
and ancient marine records in episodes of increased nutrient supply modulated by
precession cycles (Higginson et al., 2003; Swart et al., 2019). In such cases, low $\delta^{15}N_{org}$
values come with increased primary productivity and OM accumulation, just the opposite
of the relationship found in Santiurde. Alternatively, in other marine records, the shallow
water $\delta^{15}N_{org}$ signal suffered fractionation due to the liberation of bottom waters enriched
in $^{15}N_{org}$ (upwelling systems; Altabet et al., 1995). In those cases, marine productivity
increased due the liberation of nutrients stored in the sea bottom and greater OM with
relatively higher $\delta^{15}N_{org}$ signal was produced. However, the restricted palaeogeographic
setting and the sedimentary features preserved (absence of phosphatic and glauconitic
deposits) do not support the influence of upwelling currents in Santiurde.
Average $P_{EF}$ values from Santiurde are relatively depleted in P (Li and Schoonmaker,
2003), but the $P_{Ef}$ record displays a fluctuating trend with maxima at OM-rich
marls/shales (Fig. 9). Greater accumulation of P in marls/shales suggests that OM might
have increased due to enhanced marine productivity (Calvert and Pedersen, 2007).
Although Ba related indexes would not support this interpretation, it should be taken into
account that authigenic barite dissolves when bottom water oxygenation is limited
(Dymond et al., 1992; Tribovillard et al., 2006). Consequently, it is possible that the Ba
content does not reflect palaeoprodutivity ratios. $P_{EF}$ data support a relationship between
greater OM accumulation and higher palaeoproductivity, driven by the intensification of
nutrient input (Tribovillard et al., 2006; Swart et al., 2019). However, a more
comprehenssive palaeoecological study should be carried out in order to explore whether
OM fluctuations corresponded to actual variations in palaeoproductivity.
Fluctuations in the rate of dilution of OM by non-organic components can also result in
an alternation of organic-rich and organic-poor beds (Bohacs et al., 2005). In Santiurde
$C_{org}$ and phyllosilicate content show a strong positive correlation (r: 0.82; p<0.005; Fig.
12A) and covary in line with the rhytmite succession. This shows that $C_{org}$ oscillations
were not caused by variations in the rate of dilution by clays. The CaO-Al$_2$O$_3$-C$_{org}$ ternary
plot (Fig. 12C) also illustrates that the $C_{org}$/Al$_2$O$_3$ ratio is relatively constant, whereas a



higher variability is observed in the CaO/Al$_2$O$_3$ and C$_{org}$/CaO ratios. Therefore, C$_{org}$
fluctuations could have resulted from cyclic variations in the dilution rate by calcite input.
In fact, the crossplot between calcite and C$_{org}$ shows a strong negative correlation (Fig.
12B; r: -0.83; p<0.005), which is typical of dilution driven OM fluctuations (Arthur and
Dean, 1991; Beckmann et al., 2005). In order to disentangle the origin of the cyclic
sedimentation, bed thickness and duration must be taken into consideration (Einsele and
Ricken, 1991). If variations in the rate of carbonate sedimentation had been the only
process controlling organic matter dilution, while OM and clay mineral inputs stayed
constant, limestone bwould have been significantly thicker than marls/shales, which is
not the case in Santiurde (Fig. 6A). This suggests that a greater input of clay minerals
must also have occurred during the deposition of marls/shales. Moreover, samples with
higher clay mineral content and lower calcite content display greater dispersion in the C$_{org}$
vs calcite crossplot (Fig. 12B). This pattern suggests that when marl/shales were being
deposited, there might have been another factor controlling OM content, such as changes
in OM production or preservation (Bohacs et al., 2005).

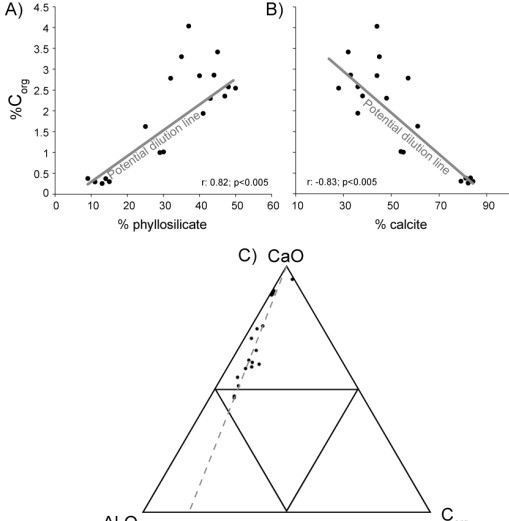

Figure 12. Crossplot of C$_{org}$ against (A) phyllosilicate and (B) calcite content. Potential dilution lines of C$_{org}$ are marked
in both graphs. C) Ca-Al-C$_{org}$ ternary plot with Santiurde samples, which follow a constant C$_{org}$/Al$_2$O$_3$.
Accordingly, the sedimentological and geochemical evidence strongly suggests that the
fluctuations in OM content found in the Santiurde rhythmite were closely related to
variations in the rate of organic-matter remineralization (preservation) as a consequence
of secular variations in seawater oxygen concentrations. Thus, the characteristics of
shales, such as well-preserved lamination, absence of burrows and the scarcity of benthic
fauna (Figs. 2 and 3), strongly suggest that the sea bottom was depleted in oxygen.
Conversely, bioturbation structures and benthic fauna are more diverse and abundant in
limestones, suggesting a better oxygenation of the seabed (Figs. 2 and 3). Changing redox
conditions can also be deduced from $\delta^{13}C_{org}$ records (Algeo and Liu, 2020). Microbial
chemoautotrophy, which is typical of oxygen-depleted environments, fixes carbon





enriched in $^{12}C$, producing lower $\delta^{13}C_{org}$ values than OM produced by photosynthetic
eukaryotic algae (Nijenhuis and Lange, 2000; Luo et al., 2014). Accordingly, minima in
$\delta^{13}C_{org}$ from OM-rich marls/shales from Santiurde are very likely related to reducing
deep-water conditions, similar to those deduced for some Pliocene Sapropels (Nijenhuis
and Lange, 2000). The strong negative correlation between $C_{org}$ content and $\delta^{13}C_{org}$ (r: -
0.945, p<0.0001) supports the close relationship between seabed oxygenation conditions
and OM preservation. This interpretation is in line with that derived from the
abovementioned C/N ratio, which also suggests that denitirification intensified during
deposition of marls/shales due to more reducing sea bottom conditions.
The interpretations above are also supported by $N_{org}$ and $\delta^{15}N_{org}$ data. Denitrification can
result in $\delta^{15}N_{org}$ isotope fractionation in poorly oxygenated conditions, as denitrification
and anaerobic ammonium oxidation reactions increase $^{15}N_{org}$ in OM (Robinson et al.,
2012). In Santiurde $\delta^{15}N_{org}$ isotopes fluctuate in line with the lithological rhytmites (Fig.
7), showing maxima at marls/shales and hence a significant negative correlation with
$\delta^{13}C_{org}$ (r: -0.70 p<0.005) and positive correlations with $C_{org}$ (r: 0.66, p<0.005) and $N_{org}$
(r: 0.73, p<0.005) content. This suggests that both isotopic signals were probably
controlled by similar environmental factors. It can therefore be concluded that $\delta^{15}N_{org}$
values increased during the accumulation of marls/shales, when bottom water
oxygenation decreased and denitrification intensified.
Pyrite and $C_{org}$ contents also show an intermediate positive correlation in Santiurde (r:
0.6, p<0.01). Pyrite might be formed during very early digenesis due to reactions between
Fe and $H_2S$. $H_2S$ is generally released into porewaters when sulphate-reducing bacteria
use sedimentary organic matter as a reducing agent and energy source (Berner, 2013).
More oxygenated conditions during the deposition of limestones could have inhibited the
formation of pyrite. Conversely, limestones present higher magnetic susceptibility values
than marls/shales, possibly associated with a greater concentration of magnetite (Fig. S2).
Magnetite could be either detrital in origin or related to postdepositional changes in redox
state, as more oxygenated conditions favour the partial replacement of pyrite with iron
oxides, such as magnetite (Lin et al., 2021).
Finally, the correlation matrix (Table 1) and the factor analysis (Fig. 11) also show a close
relationship between some redox sensitive elements (Fig. 10; V, Zn, Co, Cu, Ni), pyrite
and $C_{org}$ content (Calvert and Pedersen, 2007; Algeo and Liu, 2020). Enrichment factors
and ratios highlight a relative enrichment in redox sensitive elements throughout the
succession, which supports the general depositional model of a sea floor depleted in
oxygen (Quesada et al., 2005; Rosales et al., 2006). Trace-metal enrichment factors and
bielemental ratios associated with both sulphides and organic matter vary in line with the
lithological rhytmites and support the interpretation of alternating environmental redox
conditions. Similarly, a higher content in authigenic Barite in limestones may indicate
more oxygenated conditions.
To sum up, the multiproxy analyses shows that the higher $C_{org}$ content in marls/shales
was related to less oxygenated sea-bottom conditions, with evidence of slightly increased





palaeoproductivity. Given the close relationship between these processes and the
lithological rhythmites, it can be concluded that there must have been an orbitally driven
environmental factor that triggered fluctuations in bottom waters oxygenation and
palaeoproductivity.

### 5.3. Orbitally modulated environmental changes


Previous studies of North Iberian Pliensbachian records have demonstrated that this area
was subject to semi-arid climatic conditions, physical erosion being prevalent in the
continent and seawater being temperate (Rosales et al., 2004; Armendáriz et al., 2012,
Gómez et al., 2016; Deconinck et al., 2020). The BCB, being located close to the
boundary between the arid and humid climatic belts at  approximately 30ºN
palaeolatitude, was especially sensitive to orbitally driven climate change episodes, which
were recorded by the outer ramp hemipelagic rhytmites from Santiurde. These rhytmites
are best characterized in the stratigraphic succession by decimetre-scale calcareous
couplets, which represent precession cycles, and metre-scale bundles linked to short
eccentricity cycles. The imprint of long eccentricity cycles cannot be readily identified in
the field, but can be deduced by spectral analysis. The mutiproxy palaeoenvironmental
analysis carried out herein showed that $C_{org}$ production and preservation varied in line
with precessional cycles and was modulated by short eccentricity cycles. Although
background oxygentation of the depositional area during the BS deposition was depleted
in oxygen, the astronomically driven environmental changes ultimately determined the
occurrence of lower oxygen conditions at the seabed when marls/shales were being
accumulated and higher oxygenation conditions during limestone accumulation.

### 5.3.1. Formation of precession driven calcareous couplets


The sedimentary processes behind the formation of precession couplets can be analysed
on the basis of thickness relationships between the constituent lithologies (Einsele and
Ricken, 1991). When limy beds are thicker than marly beds, the formation of the
calcareous couplets is commonly attributed to fluctuations in either carbonate dissolution
or carbonate production. Contrarily, marls/shales are usually thicker than limestones
when periodic changes in the rate of dilution of pelagic carbonate by terrigenous
components originate the couplets. Periodic carbonate dissolution can be ruled out in
Santiurde, as there is neither macroscopic nor microscopic evidence of pervasive
carbonate dissolution and the outer carbonate ramp seabed was permanently above the
carbonate compensation depth (Bjerrum et al., 2001). In Santiurde, the low variability of
the $C_{org}/Al_2O_3$ ratio and the negative relationship between $CaCO_3$ and $C_{org}$ indicate that
fluctuations in carbonate input were an important factor in the formation of calcareous
couplets. However, the L/M ratio is close to 1 in most of the couplets (Fig. 6A).
Consequently, the formation of the Santiurde precession driven couplets most likely
responded to periodic changes in both carbonate production and carbonate dilution by
terrigenous material, increasing accumulation and preservation of $C_{org}$ when marls/shales
deposited. In fact, factor analysis points out that precession driven lithological alternation
(Fig. 11) is strongly asociated to redox sensitive variables and terrigenous proxies.



Given the generally semiarid Pliensbachian conditions deduced for the BCB (Dera et al., 2009; Deconinck et al, 2020), a climate characterized by a prolonged dry season and a short wet season can be envisaged. Dry sub-humid climates, with three to five wet months per year and a maximum degree of seasonality, produce maximum values of fluvial sediment discharge into the sea (Cecil and Dulong, 2003). Such high seasonality conditions are generally produced when the precessional configuration results in summers occurring at perihelion and winters at aphelion (Fig. 13). In Santiurde both the L/M ratio and the terrigenous content of couplets suggest that shales/marls were formed in such an astronomical configuration. Intensified monsoons during the wet season could have increased the fluvial discharges that reached periplatform areas, producing maxima of geochemical proxies associated with coarser detrital grain size, such as $Si_{Ef}$ or $Ti_{EF}$ (Fig. 9; Calvert and Pedersen, 2007). However, inorganic and organic stable isotope records do not support an increased input of fresh water or terrestrial OM when marls and shales were being deposited. Alternatively, it is also possible that the terrigenous material was transported by wind. Indeed, other studies have also related an enrichment in Si and Ti content in pelagic sediments to stronger aeolian input (Rachold and Brumsack, 2001) and increased dust production and transportation during high seasonality conditions (Woodard et al., 2011). Thus, it can be assumed that dust generation increased in the continents nearby Santiurde during the extremely dry seasons produced at precessional configurations leading to maximum seasonality. Extreme seasonality conditions may also have increased dust storms and dust input into the adjacent ocean (McGee et al., 2010). Either aeolian or fluvial, increased terrigenous input during maximum seasonality conditions may also have supplied nutrients into the ocean ($P_{EF}$), triggering organic phytoplackton blooms and organic matter production. This situation promoted greater OM accumulation and oxygen depletion in deep sea sediments (e.g. Nijenhuis and Lange, 2000; Wang, 2009; Chroustova et al., 2021). Given that the evidence of changing palaeoproductivity is scarce, it is also possible that orbitally forced mechanisms also modulated the amount of dissolved oxygen in seawater. As there are no evidences of great influence of continental water masses that could have promoted density stratification of the water column (e.g., Arthur and Dean, 1991; Chroustova et al., 2021), it is more likely that the mechanism was marine in origin. Interestingly, numerical simulations suggested that during the Late Cretaceous hothouse both precession and eccentricity cycles modulated seawater ventilation and oxygenation, driven by changes in deep ocean circulation (Sarr et al., 2022). It is therefore possible that basins that were depleted in oxygen, like the Santiurde area, were especially sensitive to orbitally forced ventilation variations. According to the model, the precessional configuration with the higher seasonality recorded the greatest oxygen depletion at intermediate and deep-water depth, producing a strong vertical oxygen gradient and seawater stratification. In Santiurde, similarly reduced vertical mixing may have occurred during the accumulation of marl/shales, which would have enhanced deep-water anoxia. Indeed, in Early Jurassic times, lower frequency orbital cycles also triggered periodic changes in the ventilation and oxygenation of bottom sediments, controlling carbonate and OM accumulation (Pieńkowski et al. 2021). Thus, the southward flow of Artic waters from the Boreal Sea into the Laurasian epicontinetal seaway favoured thermohaline circulation and the





ventilation of deep waters. However, in periods of high atmospheric $CO_2$, more sluggish
currents or stagnant conditions prevailed due to the influx of warm and saline waters from
the Tethyan area. It is possible that the early Pliensbachian BCB rhytmites recorded
similar, but probably weaker, palaeoceanographic changes at precession timescales.
Anoxic bottom water conditions allowed OM to be preserved, favoured the precipitation
of authigenic sulphides and the dissolution of Fe and Mn oxo-hidroxides (Capet et al.,
2013), and altered the organic isotopic signal (enrichment in $^{13}C_{org}$ and depletion in
$^{15}N_{org}$). Increased OM burial also resulted in a decrease in the $^{12}C$ content of inorganic
carbon dissolved in seawater (Mackensen and Schmiedl, 2019). Although the $^{13}C_{carb}$
signal found in Santiurde records this C storage fractionation, it is not possible to quantify
the digenetic imprint.

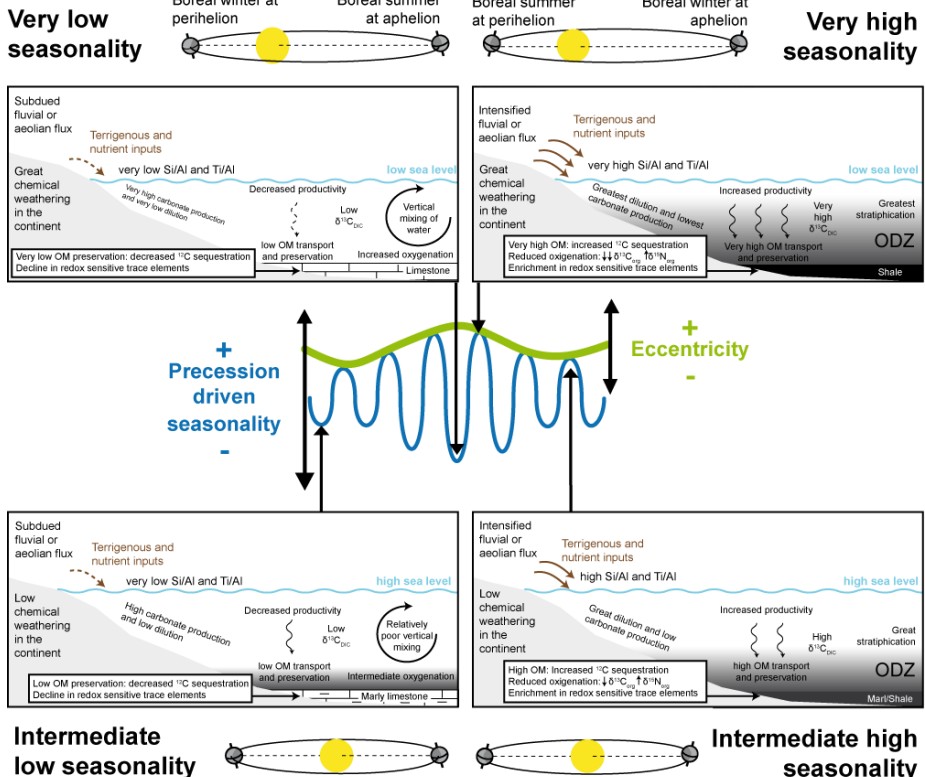

Figure 13. Orbitally tuned depositional model for the formation of the calcareous couplets and bundles from Santiurde.
Schemes on the left represent environmental conditions during precessional stages with low annual seasonality (boreal
summertime at aphelion). Schemes on the right represent environmental conditions during precessional stages with
high annual seasonality stages (boreal summertime at perihelion). The influence of maximum eccentricity is shown at
the top and that of minimum eccentricity at the bottom. DIC: Dissolved inorganic carbon. ODZ: Oxygen depleted zone.
In contrast, OM-poor limy beds accumulated during low seasonality precessional stages.
Such low seasonality conditions (mild summers and winters) resulted when summers
occurred at aphelion and winters at perihelion (Fig. 13). Mild wet and dry seasons caused
a decrease in detrital input (by wind and rivers), as well as in nutrient supply.
Consequently, organic matter production and, consequently, bottom water oxygen





consumption declined (e.g. Nijenhuis and Lange, 2000; Wang, 2009; Chroustova et al., 2021). Moreover, according to the orbitally modulated ocean circulation model (Sarr et al., 2022), low seasonality precessional stages would also have resulted in maximum values of dissolved oxygen in bottom water. These environmental conditions favoured vertical mixing of the water column, bringing oxygen to bottom waters, which allowed the oxidation of organic matter (Capet et al., 2013). Regarding carbonate components, previous studies have shown that Jurassic shelfal carbonate factories were more efficient than pelagic ooze in micrite production (Hinnov and Park 1999; Bádenas et al., 2012). It can therefore be concluded that decreased terrigenous inputs into shallow marine areas further increased shelfal carbonate mud production, surpluses being exported into deeper areas (Tucker et al., 2009; Bádenas et al., 2012). Assuming the general $\delta^{13}C_{carb}$ trend to be primary, the enrichment in $^{12}C$ of limestones could correspond to OM balance in the marine environment (Mackensen and Schmiedl, 2019). Thus, well oxygenated bottom waters allowed most of the $^{12}C$-rich OM to be oxidized before burial, decreasing the $\delta^{13}C$ of inorganic carbon dissolved in seawater.

The palaeoenvironmental model derived from the Santiurde precession couplets differs significantly from those presented by others for lower Pliesbachian successions from NW and central Europe (Fig. 1; Martinez and Dera, 2015; Hollar et al., 2023). However, it should be taken into account that these models were developed for successions accumulated in the humid climatic belt, where wet conditions prevailed throughout the year and seasonality was generally weak. In such settings, terrigenous and nutrient inputs increased at precessional configurations with higher seasonality, causing greater productivity during the wettest season and stronger vertical water mixing during the drier season. Consequently, the more calcareous OM-poor beds accumulated at high seasonality precessional stages.

### 5.3.2. Formation of eccentricity driven bundles

During an eccentricity cycle, the amplitude of precession-driven seasonality cycles is modulated by variations in the shape of the orbit of the Earth around the Sun (Berger and Loutre, 1994). At maximum eccentricity the orbit of the Earth is elliptical and, consequently, insolation changes as much as 24% in one single year, causing significantly contrasting seasonality conditions. In the northern hemisphere seasonality is maximized when summers occur at perihelion and winters at aphelion, but seasonality is minimized when winters occur at perihelion and summers at aphelion (Fig. 13). On the contrary, at minimum eccentricity the orbit of the Earth is almost circular, which results in relatively small variations in insolation between aphelion and perihelion, regardless of the precession-driven orientation of the axis of the Earth. In short, two extreme climatic situations (maximum and minimum seasonality) alternate throughout 20 kyr precession cycles at maximum eccentricity, whereas climatic conditions remain stable for longer periods at eccentricity minima.

In Santiurde the arrangement of couplets in bundles is the lithological expression of the modulation of the amplitude of precession-driven seasonality by eccentricity cycles (Fig.





2B). In the interval studied in detail, couplets C36-C37 and C41-C42, located at the
boundaries between bundles B8-B9 and B9-B10, show relatively little lithological
contrast (mals/shales alternating with marly limestones), which suggests formation at
eccentricity minima. The rest of couplets are situated in the central parts of bundles and
show a marked lithological contrast (shales alternating with limestones), which suggests
formation in the two extreme situations that occur during precession cycles at maximum
eccentricity. This amplitude modulation is also recorded by several geochemical and
mineralogical proxies, corroborating the impact of eccentricity cycles on the formation
of the rhytmite.
The fluctuations in some redox sensitive ($C_{org}$, $N_{org}$, trace elements, $\delta^{13}C_{org}$, $Mn_{EF}$) and
productivity (represented by $P_{EF}$) proxies, some of them associated with Factor 1 in the
factorial analysis (Fig. 11), display greater amplitude during eccentricity maxima. This
suggests that intensified precessional seasonality at maximum eccentricity caused an
increase in terrestrial sediment and nutrient input to the sea, which ultimately resulted in
the intesification of OM production and oxygen consumption (e.g. Nijenhuis and Lange,
2000; Wang, 2009; Chroustova et al., 2021). Precession driven variations in oceanic
currents, which controlled vertical oxygen gradient and seawater stratification, also
contributed to promoting bottom water anoxia in this orbital configration (Sarr et al.,
933 2022).

On the other hand, limy beds show significant variations in $CaCO_3$ content (Fig. 6), from
minimum values at bundle boundary couplets (e.g., 32.26% in C36L) to maximum values
in the middle part of the bundles (e.g., 88.98% in C35L). Limy beds in the central part of
bundle B9 also show the lowest content in terrigenous material and coarse-grained detrital
components (Figs. 6 and 9). Conversely, marls/shales show a significantly lower variation
in $CaCO_3$ content throughout eccentricity cycles (from 24.63 to 45.33% at C35M and
C38M, respectively), although marls in the central part of the bundle display maximum
values in terrigenous material and coarse-grained detrital indices. Therefore, eccentricity
cycles also modulated the low seasonality precessional stages, in which carbonate
accumulation was favoured (Hinnov and Park 1999; Bádenas et al., 2012). At extremely
low seasonality conditions at eccentricity maxima, continental inputs were minimal and,
consequently, so was marine OM production. At the same time, oceanic currents
intensified vertical mixing of water, favouring a well oxygenated water column and
carbonate production (Sarr et al., 2022).
Moreover, factor 2, which comprises proxies associated with dilution of carbonate by
terrigenous input, show an interesting trend in line with eccentricity bundles. Scores of
factor 2, in addition to fluctuating with the lithological alternation of calcareous couplets,
also display a larger scale trend with minimum values at eccentricity maxima and
maximum values at eccentricity minima. This trend is mainly produced by $Na_2O$ and
$^{13}C_{carb}$ (Table S5). Indeed, $Na_{EF}$ also shows a similar trend, with generally lower values
at eccentricity maxima (Fig. 9). This may record increased chemical weathering in the
continent and the release of $Na_2O$ (Marshall, 1992). This goes against the orbitally
modulated climatic model of Martinez and Dera (2015), who concluded that chemical





wathering increases during low seasonality and annually wet climates developed at
eccentricity minima. Data from Santiurde, however, suggest that the climate was drier at
eccentricity minima.

### 5.3.3. Orbitally paced sea level changes?

It is well known that, during icehouse periods, climate change driven by high-frequency
orbital cycles affects sea level due to fluctuations in the storage of water in continental
ice, causing the so called glacio-eustatic sea level changes (Steffen et al., 2010). High-
frequency sea level changes have also been deduced from many shallow marine platforms
developed in ice-free, greenhouse periods (Haq, 2014). In the absence of extensive ice
caps, sea level changes must have been caused by forcing mechanisms other than
glacioeustasy, which are still debated. The thermal expansion/contraction of water masses
causes sea level changes, but does not produce high amplitude variations (Conrad, 2013).
Fluctuations in water storage in continental areas (principally in aquifers) seems to be a
plausible forcing mechanism of decametric sea level changes during greenhouse
conditions (Wendler and Wendler, 2016). According to the aquifer-eustatic model, low
sea levels occur when large volumes of water are stored in the continents during humid
stages, whereas sea-level rises during dry epochs due to increased aquifer discharge
(Sames et al., 2020). Consequently, in a greenhouse context, orbitally driven alternations
of arid and humid periods can originate 3rd and 4th order sea level fluctuations (Wendler
and Wendler, 2016; Sames et al., 2020). Greater accumulation of $\delta^{18}O$ and $\delta^{13}C$ depleted
fresh water in the continent results in heavier $\delta^{18}O$ and $\delta^{13}C$ of inorganic carbon dissolved
in seawater, and viceversa.
Second order sea level changes occurred in Early Jurassic times in the BCB, which were
recorded by $\delta^{13}C$ in well preserved belemnites (Rosales et al., 2006). Highstand deposits
show maximum values in OM content and $\delta^{13}C$ values in belemnites, while lowstand
intervals are characterized by carbonate-rich sedimentation and lower $\delta^{13}C$ values in
belemnites. These carbon-isotope records reflect fluctuations in the $\delta^{13}C$ composition of
the inorganic carbon dissolved in seawater, which were controlled by periodic variations
in OM burial and storage of $^{12}C$ in the seabed (Quesada et al., 2005; Rosales et al., 2006).
This suggests that water stratification increased and ventilation of the seabed decreased
in highstands. Martinez and Dera (2015) showed that $\delta^{13}C$ values from Jurassic and
Lower Cretaceous perythetyan successions also recorded second and third order sea level
changes modulated by orbital cycles. According to this study, flooding of continental
areas at highstands triggered marine productivity and, consequently, seawater $\delta^{13}C$ values
increased in neritic domains.
In Santiurde, several lines of evidence suggest that short eccentricity cycles could have
modulated sea level. Factor 2 scores (which are greatly influenced by changes in
terrigenous material and $\delta^{13}C_{carb}$; see table S5) change in line with eccentricity bundles,
displaying higher values at eccentricity minima and lower values at eccentricity maxima
(Fig. 14). Average $\delta^{13}C_{carb}$, %CaCO₃ and Ti$_{EF}$ values per couplet show high values at
eccentricity minima. Average C$_{org}$ and N$_{org}$ values per couplet also fluctuate in line with





eccentricity bundles, showing maximum (or minimum) values in the intervals that
correspond to low (or high) eccentricity configurations. This may indicate that the average
OM content per precessional stage was higher at eccentricity minima, although shales at
eccentricity maxima recorded maximum OM values. Using the abovementioned models,
it can be postulated that low sea levels may have occurred during eccentricity maxima.
Lowstand deposits recorded the highest and probably coarsest terrigenous inputs ($Ti_{EF}$;
Olde et al., 2015), but also the most calcareous sedimentation due to platform
progradation. A lower sea level would have facilitated seawater ventilation and OM
degradation at eccentricity scale. However, ventilation at maximum eccentricity
decreased when precession-driven seasonality increased, which temporarily enhanced
OM production and preservation, and caused the accumulation of shales on the seabed.
Similarly, a higher sea level at eccentricity minima could have decreased bottom water
ventilation, contributing to OM preservation. These conditions promoted OM
accumulation even if terrigenous and nutrient inputs were not high when shales deposited.

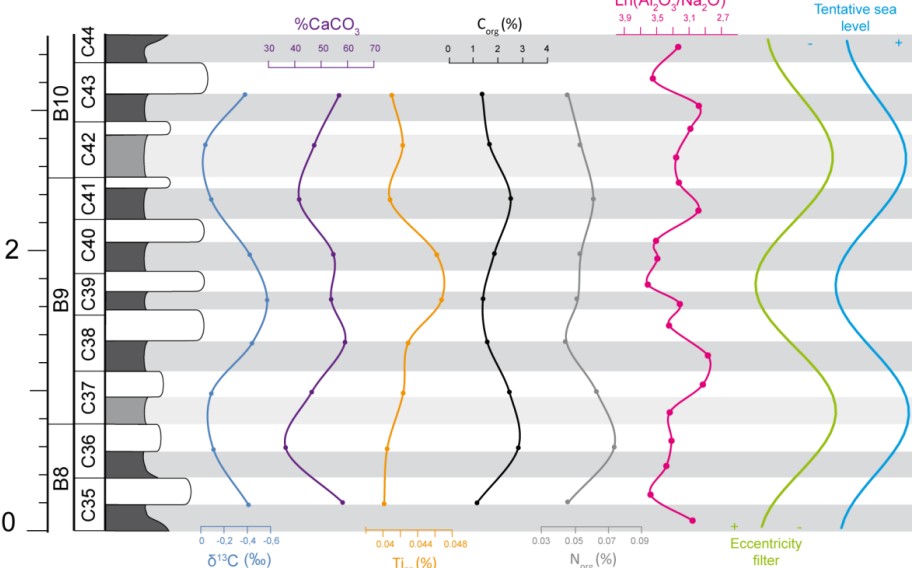


Figure 14. Lithological log of the Santiurde interval studied in detail, showing the average value per couplet of $\delta^{13}C_{carb}$,
%$CaCO_3$, $Ti_{EF}$, $C_{org}$ and $N_{org}$. The palaeoweathering index $Ln(Al_2O_3/Na_2O)$ of all beds, the short eccentricity colour
filter output (Fig. 5) and a tentative sea level curve are also shown.
Minima of $Na_{EF}$ at high eccentricity lowstands (Fig. 8) suggest that the climate may have
been more humid than during low eccentricity highstands. The $Ln(Al_2O_3/Na_2O)$ index is
a palaeoweathering index based on a statistical model of linear compositional and
weathering trends (Von Eynatten et al., 2003). This index is especially recommended for
rocks with a high percentage of biogenic carbonate (Montero-Serrano et al., 2015), such
as those from Santiurde. $Ln(Al_2O_3/Na_2O)$ values in Santiurde show a gradual trend in line
with eccentricity bundles (Fig. 14). Maximum values, which indicate greater chemical
weathering in the continent, are recorded at eccentricity maxima. This configuration
agrees with the aquifer-eustatic sea level model, in which humid climates result in
increased fresh water storage in the continent and lower sea levels, whereas aquifers are





emptied in drier periods and sea-level rises (Wendler and Wendler, 2016). Jurassic sea
level changes deduced from shallower areas from the Iberian basin were also associated
with orbitally paced aquifer-eustatism (Sequero et al., 2017; Val et al., 2017).

### 5.3.4. Comparison with orbital forcing during Mesozoic OAEs

Four Lower Jurassic BS levels occur in the BCB and the Asturian basin (Borrego et al.,
1996; Rosales et al., 2006). The lower Toarcian BS correlates with the globally recorded
early Toarcian Oceanic Anoxic Event (T-OAE; Jenkyns and Clayton, 1986; Hesselbo et
al., 2000; Rosales et al., 2006), which was related to a perturbation in the Earth's climate
originated by an abrupt addition of $^{12}$C into the carbon cycle. Many studies have
previously demonstrated the influence of orbital forcing on the T-OAE in western,
southern and northem Tethys areas (Huang and Hesselbo, 2014; Boulila and Hinnov,
2017, Boulila et al., 2019). These studies revealed the general prevalence of 405-kyr
eccentricity cycles in lower Jurassic records, along with a strong expression of both
precession and obliquity cycles, although the influence of the latter only increased during
the anoxic event. The palaeoenvironmental changes driven by obliquity cycles produced
variations in productivity, seabed oxygenation and/or OM origin during the T-OAE (Suan
et al., 2015). The shift in astronomical forcing during the T-OAE has also been linked
with the lengthening of the terrestrial productivity season due to increaseses in global
temperatures and humidity (Boulila and Hinnov, 2017; Boulila et al., 2019).
In Santiurde, the influence of eccentricity and precession cycles prevailed during the
formation of the Pliensbachian BS1, with little or no evidence of obliquity forcing.
Interestingly, however, precession cycles also modulated the palaeoenvironmental
changes (continental weathering, oceanic productivity and redox conditions) that
occurred during other Mesozoic OAEs associated with the release of greenhouse gases,
such as the Cretaceous OAE 1a and 1b events (Giogiorni et al 2015; Benamara et al.,
2020). It can therefore be concluded that the Pliensbachian BS1 of the BCB shows greater
similarities with Cretaceous OAEs than with the Toarcian OAE. However, it should be
noted that most of the astrochronological studies of the Early Jurassic, including those
focused on orbital forcing on the T-OAE, were previously focused on successions located
at higher latitudes than Santiurde (Suan et al., 2015; Martinez and Dera, 2015; Boulila
and Hinnov, 2017; Storm et al., 2020). It is possible that, similar to the eccentricity
modulated precessional depositional model, climatic belts determined the response of the
sedimentary environment to similar climatic forcings.

### 6. Conclusions

Lower Pliensbachian organic-rich calcareous rhytmites from the BCB are the expression
of periodic environmental variations that occurred in the Milankovitch-cycle band. The
cyclostratigraphic analysis of rock colour and magnetic susceptibility data series showed
that calcareous couplets represent precession cycles, whereas thicker bundles record short
eccentricity cycles; the effect of long-eccentricity cycles, despite not being well expressed
in the field, was also identified.





The integrated sedimentological, mineralogical and geochemical analysis of a short eccentricity bundle allowed the identification of the environmental factors that governed the formation of the rhytmite, as well as the assessment of diagenetic overprinting. Most of the compositional parameters record primary characteristics related to the formation of the calcareous rhytmites, but inorganic stable isotope records and the distribution of several trace elements may have been somewhat affected by diagenesis during burial. However, the results allowed the definition of an original orbitally modulated depositional model which provides new insight into the formation of lower Pliensbachian organic-rich calcareous rhytmites.

The formation of precessional calcareous couplets was regulated by variations in carbonate productivity and in dilution by terrigenous supplies. Thus, organic-rich marls and shales deposited during precessional configurations which led to marked annual seasonality (boreal summer at perihelion and winter at aphelion). Increased seasonal rainfall on land and terrigenous input (by rivers or wind) to marine areas boosted organic productivity in surface waters. Increased accumulation of organic matter on the seabed eventually caused poorly oxygenated bottom waters. Deep-sea desoxygenation and seawater stratification were enhanced due to changes in ocean circulation. Conversely, limy beds were formed when seasonality was minimal (boreal winter at perihelion and summer at aphelion). The consequent decrease in terrigenous inputs favoured a greater production and basinward exportation of carbonate sediment in shallow marine areas. A lower production of OM and increased vertical seawater mixing due to changes in oceanic currents, resulted in the oxidation of organic matter in the deepest environments.

In addition, several proxies support that the precessional contrast between the intensity of seasonally controlled environmental factors, such as terrigenous input and oxygenation of bottom sea water, diminished when the Earth's orbit was circular (minimum eccentricity) and increased when it was more elliptical (maximum eccentricity). The available data further suggest that short-term sea level changes may have occurred in line with short eccentricity cycles (higher sea level at eccentricity minima), probably through orbitally modulated aquifer-eustasy.

The comparison with Lower Jurassic successions from other areas suggests that palaeolatitudinal climatic belts played a significat role in the response of the environment to astronomically forced climate-change episodes.

## 7. Competing interests

The contact author has declared that none of the authors has any competing interests

## 8. Acknowledgements

Research funded by projects PID2019-105670GB-I00/AEI/10.13039/501100011033 of the Spanish Government (MCIN/AEI) and by the Consolidated Research Group IT602-22 of the Basque Government. NM-B is grateful for post-doctoral specialization grants DOCREC19/35 and ESPDOC21/49 from the University of the Basque Country





(UPV/EHU) and a Margarita Salas contract (MARSA22/05) of the Spanish Government
with Next Generation funds from the European Union. Thanks are due to Carl Sheaver
for his language corrections.

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
