# Peer review of "of a Pliensbachian (Lower Jurassic) black shale in northern Iberia"

_Climate of the Past, 2024_

## Referee Comment (RC2)

Dear editor and author,

This paper is a solid work on the climate control on a Lower Jurassic hemipelagic succession in the Basque-Cantabrian Basin that contain interesting approaches to understand factors controlling its accumulation. Data, interpretations and discussion are very well organized (although some parts are not balanced: see comment 20; and the discussion is quite long and complex). Without a doubt, the paper deserves to be published. However, concerning descriptions (and related interpretations and discussions) four main aspects require to be deeply explained:
- hemipelagic character of the successions (see manly comments 1, 6);
- significance of color (see comments 10, 16, 18) and MS data (see comments 17, 18);
- criteria for definition of couplets (precession cycles) and bundles (eccentricity cycle) (see comment 15);
- characterization of the black shale package as a whole (see comments 7, 13, 26).

Other changes are suggested in order to state clear some concepts and descriptions.

**Introduction**

1. Pelagic rhythmites are presented as one of the key sedimentary successions recording orbital controlled climate changes (first paragraph). However, the studied succession is hemipelagic. It would be interesting to include: 1st) a brief definition of the term hemipelagic in the context of the studied BCB; 2nd) a brief explanation (and references) on the role of orbital-induce climate variations on this particular kind of sediments, compared to the pelagic ones.

**Geological setting**

2. Lines 83-84: "which connected the Boreal Sea with the southern Tethyan Ocean". Better: "which connected the Boreal Sea with the northwestern Tethyan Ocean".

3. Line 87: "source area was located in the semiarid belt". What do you mean thin "source area", emerged land?, shallow platform carbonate source area? Please, explain better.

4. Specify if the distribution of the humid and semi-arid zones was stable for the entire Early Jurassic.

5. Use in Fig. 1, Early/Lower Jurassic instead of Lias.

6. Line 104: "Pliensbachian (192.9–184.2 Ma) hemipelagic successions of the BCB..". I suggest deleting the time duration: I suppose the studied succession has not been time-calibrated so accurately. The sedimentary environment of the successions requires a deep explanation. Notice the term "outer ramp" appears for the first time in the discussion (line 778). See also lines 677-679 "restricted paleogeographic setting"). Revise also lines 841-843 ("basins depleted in oxygen": be careful, it sounds like a circular reasoning).

7. Line 106: use "packages of alternating black shales and limestones/marly limstones" instead of "black shale intervals". it is important to state clear these black shales do not include only shales but also intercalated limestone/marly limestones. I think the word Interval has a time connotation.

8. Lines 130-132: "and 1 km north-west of a coeval section studied by others at the train station in the same locality…with which a bed by-bed correlation can be readily carried out.". This sentence is more appropriate for the discussion (see also comment 26). In any case, it requires a deep explanation of how this correlation was made, without (I suppose) lateral continuity of outcrops.

9. Lines 132-137: Please state clearer the location and thickness of the studied succession. As far I understand the studied succession is 22.5 m thick and includes: the uppermost 2.5 m of the Puerto Pozazal Formation and the lowermost 20 m of the Camino Formation (including the first x-thick black shale package of this unit). However, in line 140 "30.40 m thick" is mentioned.

**Materials and methods**

10. The average color of samples is used for cyclostratigraphic (spectral) analysis. However, there is not any analysis to elucidate the sedimentary vs. diagenetic significance of this feature.

11. Thin sections are mentioned in results, but not included here.

12. Please, explain the lithology of the studied bundle and samples: line 167: fifty-seven samples, include also here the values of x samples/bed; line 177: central part of each bed, include here also the total number of samples.

**Results**

13. Lines 209-210: Concerning lithological terms, "limestones or marly limestones" and "marls or shales". Do you have calcimetric analysis of the entire succession to differentiated these lithologies?. Concerning the term "shale", please see previous comment 7. The black shale package has to be presented.

14. Description of lithologies and texture. In Fig. 2 (log), marly limestones of limestones with different texture are not drawn. I suggest to draw them. Also state clear the description of each lithology separately (also limestones and marly limestones; do they have bioturbation?) and then compare their main differences.

15. Lines 236-244 on couplets and bundles. This paragraph has to be separated in a subsection. The criteria for differentiating couplets are unclear: why the couples marl/shale to limestone/marly limestone (and not at the contrary)?; the "lithological contrast" for bundles is also very unclear (see also comment 13 on carbonate content

of the entire succession). Do you see significant features at the boundaries of couplets or bundles or any trends withing couplets or bundles?.

16. Color trends: lines 244-258 "The variations in colour values are more significant in the central couplets of bundles than at bundle boundaries. This suggests that, as shown in previous studies… colour values are representative of the carbonate content of the samples.". See previous comment 15 on "lithological contrast" for bundles (not well explained" and also comment 10 (significance of color). To use the similar trend in color and carbonate content in C35 to C44 as supporting criterion, it is necessary to discuss there was not a diagenetic imprint in both color and carbonate content.

17. Did you perform analysis of susceptibility-temperature (k-t) curves to know the type and abundance of magnetic minerals? The following sentence is not clear (as far I understand you interpret the presence of ferromagnetic minerals indirectly): Lines 264 "The MS of hemipelagic deposits is commonly determined by their paramagnetic components (mostly detrital clays; Kodama and Hinnov, 2015). However, in Santiurde this parameter does not show a great correlation with colour (r: 0.48, p<0.001, all section; Fig. S1) or calcium carbonate (r: 0.36, p<0.001, between C35 and C44; Fig. S1). Therefore, the Santiurde relationship suggests that the MS signal is more likely controlled by ferromagnetic minerals, such as magnetite (Fig. S2)." Revise also lines 750-755.

18. Spectral analysis of MS data (lines 283-285). MS data do not correlate with color and carbonate content; however, their spectral analysis corroborate the results of the spectra analysis of color. Please, explain this apparent contradiction.

19. Lines 310-311. "In general, %CaCO3 fluctuates in line with lithology, limestones and marly limestones (average: 66.36%) being richer than marls and shales (average: 34.86%).". What do you mean? In fact, carbonate content is the criterion to differentiate these lithologies.

20. In 4.2. Detailed analysis of Bundle 9 (C35-C44 interval), pure descriptions are included in 4.2.1 to 4.2.4; however, 4.2.5 and 4.2.6 contain interpretation/discussion of the results, including the interpretation of oxic/anoxic conditions of the different lithologies (without any reference to the other results). This imbalance should be corrected.

**Discussion**

21. Line 459. "Origin of inorganic sedimentary fluctuations". I suggest deleting "inorganic". This term is obscure.

22. Lines 470-472 (secondary cements..), line 474 (bed geometry): these descriptions should be explained also in Results.

23. Lines 476-478. "Quite the opposite, the characteristics of the beds are continuous for more than 1 km between the Santiurde motorway and railway sections". See comment 8.

24. Lines 485-487: "In general, the diagenetic characteristics observed in the Santiurde rhytmites are typical of processes related to organic matter decay during burial (Rosales et al., 2001). This sentence is not informative. Please explain in which way.

25. Lines 488-493 about periodicities. Do you have data on the time span of the studied succession to compare with your results? I would be interesting to know how many cycles are then represented in the entire succession and BS package.

26. The discussion lacks a proper explanation of the BS package as a whole (how many precession or eccentricity cycles includes, what short- and long-term factors controlled its accumulation.

Regards,
Beatriz Bádenas

---

## Author Comment (AC2)

**Reply to Reviewer #2 (Sietske Batenburg)**

Review Martinez-Braceras, CP, 2024

The manuscript on 'Orbitally forced environmental changes during the accumulation of a Pliensbachian (Lower Jurassic) black shale in northern Iberia' by Martinez-Braceras and co-authors investigates factors driving sedimentary rhythms in a Pliensbachian black shale interval. This is a timely approach, as sedimentary rhythms in Mesozoic successions are commonly used to construct astronomically tuned time scales, although the exact mechanisms driving lithological alternations (especially on the precessional scale) are often insufficiently understood. The authors present a multi-proxy study to discuss a suite of processes in detail, shedding light on the periodic nature of regional anoxia that resulted in the deposition of organic matter. This is a very thorough study and merits publication in Climate of the Past if some minor points can be addressed.

As the stratigraphic interval has been studied previously in the same region, it would be relevant to report any independent age information. This would include biostratigraphic, magnetostratigraphic and chemostratigraphic events, and if available (correlation to) radioisotopic ages. If no age information is available, the authors should make it clear that the interpretation of orbital forcing of the sedimentary rhythms is based solely on the cycle hierarchy.

RESPONSE: AGREE. We will improve the presentation of the chronostratigraphic data available for the studied interval, which is based on ammonite zones (Braga et al., 1988) and calcareous nannofossil zones (Fraguas et al., 2015). Unfortunately, the biostratigraphic age data does not provide the resolution needed to assess accurately the chronology of this relatively short succession at astronomical timescales.

The photograph of the section (Fig 2) shows very clear banding patterns, with individual lithological alternations varying in intensity and showing grouping in bundles. The time series analyses of the colour signal show periodicities at 6.6, 1.67, 1 and 0.37 m, where the 'intermediate' periodicities have some peculiarities. The periodicity at 1.6 m is not very strongly present in the time series analysis, whereas it is prominent in the log and view of the section. The periodicity at 1 m seems a bit different from what would be expected with a cycle hierarchy of eccentricity-modulated precession and obliquity (20:5:2:1). The longest periodicity that is strongly present in the spectral results likely reflects the influence of 405 kyr cycles, but its expression in the section is not clear. It would be good if the authors could comment on the reasons for the seeming discrepancies in the lithological patterns and the spectral analysis result.

RESPONSE: AGREE. It is true that calcareous couplets of 32-42 cm and bundles of approximately 1.65 m constitute the clearest bed arrangement in the outcrop, which correspond to the expression of precession (20 kyr) and short (100 kyr) eccentricity cycles, respectively. However, long eccentricity cycles (405 kyr) are also expressed in the section by the alternation of 3.3-m-thick intervals in which two successive short eccentricity bundles are clearly recorded (e.g., B9 and B10), and 3.3-m-thick intervals in which another two short eccentricity bundles are not so clearly defined (e.g., the underlying B7-B8 and the overlying B11-B12). The former intervals are interpreted as

long eccentricity minima, the latter as maxima. We will add a new figure (Fig. S1A), where the record of 405 kyr cycles can be readily appreciated. As the physical expression of short eccentricity bundles is subdued at long eccentricity maxima, the power of the former is relatively weak in the time series analysis. This explains the relatively low intensity of the intermediate, 1.6-m-thick periodicity in the spectra.

The time series analyses of the colour signal show a significant 1-m-thick cyclicity, which could not be identified visually in the outcrop. Based on an average duration of 20 kyr for each precession-driven couplet, this intermediate periodicity would represent 53 kyr, which could correspond to the 52.8 kyr term of obliquity (p+S6). This 1-m-thick cyclicity is not recorded in the MS spectra, but a less significant periodicity of 65 cm is identified. Based on the average duration of 20 kyr for each calcareous couplet, this intermediate MS periodicity would represent 35 kyr. The mean duration estimated from both proxies is 44 kyr, suggesting that they might be the result of obliquity. However, as the results of the two datasets are not fully coherent, these periodicities (marked as O? in Fig. 4) were not further considered in our discussion about the orbitally modulated environmental evolution of the area.

A related point is that the periodicities detected through time series analyses are consistently shorter than those observed. The number of interpreted bundles (14) in a 31 m interval suggest that the imprint of short eccentricity actually resulted in a 2.2 m cycle rather than a 1.6 m one. The 6.6 m periodicity has a much stronger peak in the spectrum but based on the number of individual alternations (62), it is only present 3 times in the studied section, and has a length of approx. 10 m rather than 6.6 m. The individual alternations have an average thickness of 31/62=0.5 m rather than 0.37 m. I do not understand the origin of this discrepancy. It has been observed that the highest amplitude cycles may have higher sedimentation rates (in the absence of dissolution) and these thicker couplets may dominate the time series analyses results. But here, the time series results indicate shorter periodicities. I would recommend the authors to evaluate other power spectra methods, to see whether these give similar results, and to report on the imposed settings more elaborately. Also, it would be interesting to generate a power spectrum for $CaCO_3$ in the studied high-resolution interval, to see whether the statistically identified periodicity driving the limestone marl alternations corresponds to the observed thickness of the alternations.

RESPONSE: AGREE. We apologize for conveying misleading information in our original manuscript. As pointed out by the other reviewer (Beatriz Badenas), the thickness of the studied section was not clearly presented. We analysed a 30.4-m-thick section in the outcrop, but the lowermost 7.9 m were excluded from the cyclostratigraphic analysis because of poor exposure. Thus, in our original manuscript the top of the stratigraphic log was located at 30.40 m, but it started at 7.9 m. This means that the studied succession is actually 22.5 m thick. In order to present this information more clearly, the bottom of the studied succession will be established at 0 m and the top at 22.5 m in the revised manuscript (Figs. 2, 4, 6, 7, 9, 10, 11, 14 and the supplementary material will be modified accordingly).

Taking into account the thickness of 22.5 m of the studied section, the periodicities detected in the time series analyses are consistent with those observed in the succession.

The 13.8 bundles identified in the 22.5 m thick succession present an average thickness of 1.63 m, which matches the 1.67 m periodicity deduced from the spectral analyses. Similarly, the 62 calcareous couplets display an average length of 36 cm, which is practically identical to the 37 cm peak and the average of the 32-42 cm band identified in the spectral analyses. Additionally, the $2\pi$-MTM power spectrum of the $CaCO_3$ data of the bundle (B9) studied in detail (see figure below) identifies a significant peak between 31-42 cm (mean value of 36.7 cm), which also corresponds to the thickness of the limestone-marl couplets.

[Figure]

It would be good to indicate how couplets and eccentricity bundles are defined here precisely, along the lines of: 'The term couplet, as used here, refers to a lithological alternation, consisting of a resistant limestone bed with a more weathered marl or shale bed, starting at the base of the marl or shale. These couplets vary in their amount of lithological contrast between the marl/shale and the limestone. The variations in lithological contrast result in a grouping into bundles of five (four to six) couplets, counting from the base of the lightest coloured marls, reflecting the least lithological contrast with their bounding limestones.'

RESPONSE: AGREE. Following also the other reviewer's comment, a specific section (4.1.2. Bed arrangement) will be added in the revised manuscript.

The L/M ratio being close to 1 is taken as indication that carbonate productivity and dilution varied hand in hand. Besides this ratio, it would be interesting to plot the thickness of the couplets and the thickness of the individual beds, to see whether for example thicker couplets coincide with thicker limestones or not.

RESPONSE: DISAGREE. The interpretation that carbonate productivity and dilution varied together is not only based on the L/M thickness ratio, but also on other geochemical characteristics obtained from the interval (bundle B9 at 12.4-15.95 m) studied in detail. Such a comprehensive dataset is not available for the entire section (0-22.5 m), which means that it is not possible to deduce the variations in environmental conditions over time based solely on L/M thickness ratios. However, following the reviewer's suggestion,

we calculated the L/M ration of the entire succession (0-22.5 m). The average L/M ratio of 1.08 obtained from the couplets of the entire Black Shale interval (C10 to C45 at 2.5-16.4 m) is similar to that obtained in bundle B9. The figure below shows the thickness and L/M ratio of all the couplets. There is not a clear or repetitive trend between the thickness of couplets and their L/M ratio. Consequently, we consider that this information does not contribute significantly to the main scope of our study.

[Figure]

A range of geochemical methods is applied to carefully investigate the factors controlling the production and preservation of organic matter. Changes in P-EF seem to suggest elevated productivity in the dark levels, but this is contrasted by δ15Norg, δ13Corg and Ba-EF, which are explained to suggest lower productivity. I wonder if the authors can comment on whether, instead of increased productivity, enhanced preservation would be sufficient to explain the observed patterns.

RESPONSE: AGREE. The fact that changes in the preservation of organic matter constituted the main factor that controlled the $C_{org}$ content will be more clearly explained in the revised manuscript. The multiproxy analysis ($\delta^{15}$Norg, $\delta^{13}$Corg, trace elements, mineralogy and sedimentology) shows that the higher $C_{org}$ content in marls/shales was related to less oxygenated sea-floor conditions, which enhanced the preservation potential of organic matter. The $P_{EF}$ record suggests that the production of organic matter may also have increased during the formation of marls/shales, but this signal is not coherent throughout the studied interval. Given the close relationship between these processes and the lithological rhythmites, it can be concluded that there must have been an orbitally driven environmental factor that triggered fluctuations in bottom water oxygenation and, possibly, palaeoproductivity.

The $\delta^{13}$C changes are addressed in many parts of the manuscript, and perhaps the readability would benefit from grouping all information about δ13C together, or a paragraph summarizing it.

RESPONSE: DISAGREE. It would be rather difficult to concentrate the discussion about $\delta^{13}C_{carb}$ data in one single section, because it is used to asses both the diagenetic overprinting and the orbitally modulated environmental changes (sections 5.1 and 5.3). Moreover, $\delta^{13}C_{carb}$ results were not more significant than other geochemical or mineralogical proxies (none of which is discussed in specific sections) for the development of the cyclic sedimentation model. In this regard, the only exception is the content in organic matter (specifically addressed in section 5.2), but this is due to the fact that the great organic matter content is the main feature that characterizes the Basque-Cantabrian Lower Jurassic successions and why we selected this interval for our study.

Minor points

L 21: change 'involved processes' to 'processes involved'
RESPONSE: AGREE. The manuscript will be revised accordingly.

L 22: change 'The study' to 'This study'
RESPONSE: AGREE. The manuscript will be revised accordingly.

L 23: change 'black shales' to 'black shale intervals', change 'revealed' to 'reveals'
RESPONSE: AGREE. The manuscript will be revised accordingly.

L 25: the phrase 'with the prevalence of precession, short eccentricity and long eccentricity cycles' could be replaced by something a like: 'and were likely driven by eccentricity-modulated precession' to be more precise
RESPONSE: AGREE. The manuscript will be revised accordingly.

L 32: the comma has to be deleted to understand what the active verbs are in the sentence
RESPONSE: AGREE. The manuscript will be revised accordingly.

L 34: change waters to water
RESPONSE: AGREE. The manuscript will be revised accordingly.

L 36: change maximum to maximal
RESPONSE: AGREE. The manuscript will be revised accordingly.

L 37: typo in diminished
RESPONSE: AGREE. The manuscript will be revised accordingly.

L 38: delete seawater
RESPONSE: AGREE. The manuscript will be revised accordingly.

L 39: add and before contributed

RESPONSE: PARTLY AGREE. This sentence will be rephrased in another way in order to improve its meaning.

L 40: change exportation to export

RESPONSE: AGREE. The manuscript will be revised accordingly.

L 43: change seawaters to water

RESPONSE: AGREE. The manuscript will be revised accordingly.

L 46: change orbital to orbitally

RESPONSE: AGREE. The manuscript will be revised accordingly.

L 50: change few to a few

RESPONSE: AGREE. The manuscript will be revised accordingly.

L 51: add 'and temporal' after latitudinal

RESPONSE: AGREE. The manuscript will be revised accordingly.

L 56: move 'erode the seabed' and 'or' to before 'interrupt'

RESPONSE: AGREE. The manuscript will be revised accordingly.

L 57: delete ', sedimentation'

RESPONSE: AGREE. The manuscript will be revised accordingly.

L 80: delete on

RESPONSE: AGREE. The manuscript will be revised accordingly.

L 82: replace Armorican by 'the Armorican Massif'

RESPONSE: AGREE. The manuscript will be revised accordingly.

L 83: replace being part of by 'within'

RESPONSE: AGREE. The manuscript will be revised accordingly.

L 99: delete was

RESPONSE: AGREE. The manuscript will be revised accordingly.

L 106: Here and in other occasion: I recommend avoiding the abbreviation BS which is in English is commonly used to refer to bullshit.

RESPONSE: AGREE. BS will be replaced with BSI throughout the text to refer to "Black shale interval".

L 111: replace United Kindom with 'the United Kingdom'

RESPONSE: AGREE. The manuscript will be revised accordingly.

L 116: put in before inland

RESPONSE: AGREE. The manuscript will be revised accordingly.

L 120: replace on by of

RESPONSE: AGREE. The manuscript will be revised accordingly.

L 208: weather resistant and weather recessive does not sound correct. You could delete 'weather' or you could explain that the beds are either resistant or susceptible to weathering.

RESPONSE: AGREE. The manuscript will be revised accordingly.

L 216: as L 208

RESPONSE: AGREE. The manuscript will be revised accordingly.

L 218: add the before marls

RESPONSE: AGREE. The manuscript will be revised accordingly.

L 220: add and before trace

RESPONSE: DISAGREE. The information in brackets is a list of characteristics, separated by semicolons.

L 234: delete weather (2x)

RESPONSE: AGREE. The manuscript will be revised accordingly.

L 277: replace which peaks at by with a main periodicity of

RESPONSE: AGREE. The manuscript will be revised accordingly.

L 288: to help the reader, please mention the width of the filters in the main text, expressed as periodicities. Consider explaining why the bandwidths of the two filters are very different (half of the centre frequency vs one fourth of the centre frequency).

RESPONSE: AGREE. We agree that there was no coherence between the bandwidths of both filter outputs. Consequently, a new filter output will be extracted for the intermediate frequency and included in the new Figure 5, with a bandwidth close to one fourth of the centre frequency (similar to that used in the filter of the short periodicity).

L 293: I suggest to replace the word chronostratigraphy by cyclostratigraphic interpretation. Ideally, an integrated chronostratigraphy would include information from bioevents, magnetostratigraphy, chemostratigraphy, radioisotopic dating, etc.

RESPONSE: PARTLY AGREE. The chronostratigraphic information refers to the Jamesoni biozone of the Pliensbachian stage (obtained from Quesada et al., 2005, and Rosales et al., 2006), depicted on the left of the stratigraphic log. In Figure 4 we do not present our cyclostratigraphic interpretation of the Santiurde section, but only the results of our spectral analysis. In order to clarify this misunderstanding, we will add the reference for the chronostratigraphic data in the revised manuscript.

L 299: add cycles after m, replace corresponds by correspond

RESPONSE: AGREE. The manuscript will be revised accordingly.L 311: add 'in CaCO3' after richer

RESPONSE: AGREE. The manuscript will be revised accordingly.

L 313: delete counterpart

RESPONSE: AGREE. The manuscript will be revised accordingly.

L 340: add with before maximum

RESPONSE: AGREE. The manuscript will be revised accordingly.

L 375: replace 'the amplitude of the oscillations' by 'amplitude of variability'

RESPONSE: AGREE. The manuscript will be revised accordingly.

L 422: replace seawater by water, and 'concentration was' by 'concentrations were'

RESPONSE: AGREE. The manuscript will be revised accordingly.

L 434: replace whose by which, add of before their

RESPONSE: AGREE. The manuscript will be revised accordingly.

L 557: add with before that

RESPONSE: AGREE. The manuscript will be revised accordingly.

L 565: replace records by record

RESPONSE: AGREE. The manuscript will be revised accordingly.

L 567: replace alternation by alternations

RESPONSE: AGREE. The manuscript will be revised accordingly.

L 577: replace if by when

RESPONSE: AGREE. The manuscript will be revised accordingly.

L 606: add than after higher

RESPONSE: AGREE. The manuscript will be revised accordingly.

L 625: replace indurate by indurated

RESPONSE: AGREE. The manuscript will be revised accordingly.

L 632: replace indurate by indurated

RESPONSE: AGREE. The manuscript will be revised accordingly.

L 655: replace originate by lead to

RESPONSE: AGREE. The manuscript will be revised accordingly.

L 660: add strength of before biological pump

RESPONSE: AGREE. The manuscript will be revised accordingly.

L 664: replace distortions by alterations

RESPONSE: AGREE. The manuscript will be revised accordingly.

L 672: replace come by coincide

RESPONSE: AGREE. The manuscript will be revised accordingly.

L 676: replace 'and greater OM with relatively higher' by 'and more OM with a relatively higher'

RESPONSE: AGREE. The manuscript will be revised accordingly.

L 681: the PEf record actually does not always have its maxima in black shales. Perhaps mention the P concentrations themselves to strengthen the observation.

RESPONSE: AGREE. The manuscript will be revised accordingly.

L 706: typo in would

RESPONSE: AGREE. The manuscript will be revised accordingly.

L 721: replace sea bottom by either sea floor or bottom water

RESPONSE: AGREE. The manuscript will be revised accordingly.

L 766: replace bottom by floor

RESPONSE: AGREE. The manuscript will be revised accordingly.

L 769: replace waters by water

RESPONSE: AGREE. The manuscript will be revised accordingly.

L 785: typo in oxygenation

RESPONSE: AGREE. The manuscript will be revised accordingly.

L 835: replace are no evidences by is no evidence

RESPONSE: AGREE. The manuscript will be revised accordingly.

L 844: replace depth by depths

RESPONSE: AGREE. The manuscript will be revised accordingly.

L 862: typo in diagenetic

RESPONSE: AGREE. The manuscript will be revised accordingly.

L 885: add the before OM

RESPONSE: AGREE. The manuscript will be revised accordingly.

L 1076: replace supplies by supply

RESPONSE: AGREE. The manuscript will be revised accordingly.

L 1090: delete sea

RESPONSE: AGREE. The manuscript will be revised accordingly.

L 1096: typo in significant

RESPONSE: AGREE. The manuscript will be revised accordingly.

Figure 4: replace 'Relief in the outcrop' by 'weathering profile'. Consider reverting the colour axis back, so that the peaks coincide with protruding beds and are more easily compared with the log.

RESPONSE: AGREE. Figure 4 will be modified accordingly.

Figure 13: This is an excellent summary of your findings and the different orbital configurations are explained well. Instead of Ti/Al and Si/Al, I recommend using the enrichment factors that you use in the text and other figures. As the role of productivity is not so well constrained, I recommend using a question mark after the claim of increased productivity. Similarly, you could consider including only low/high OM preservation in the text within the figure (rather than including the transport)

RESPONSE: AGREE. Figure 13 will be modified accordingly.

---

## Author Response (AR1)

Dear editor(s) of **Climate of the Past**,

I hereby submit the revised version of manuscript CP-2024-13, entitled "**Orbitally forced environmental changes during the accumulation of a Pliensbachian (Lower Jurassic) black shale in northern Iberia**", which has been approved by all co-authores (Naroa Martínez-Braceras, Aitor Payros, Jaume Dinarès-Turell, Idoia Rosales, Javier Arostegi and Roi Silva-Casal).

We thank the reviewers (Beatriz Badenas and Sietske Batenburg) and editor (Gerilyn Soreghan) for their constructive comments. As we are largely in agreement with most of the suggestions, the manuscript has been modified accordingly. We think that all these changes have clarified and improved the original manuscript considerably in line with the reviewers' recommendations. We therefore hope that now you will find it better suited for publication in **Climate of the Past**.

Point by point responses to the reviewers' and editor´s comments are given below. The corrections made in the revised manuscript can be seen in the files that we have uploaded: (1) A version of the revised manuscript where all modifications are highlighted, (2) the new, clean version of the revised manuscript and (3) the revised supplementary material.

Thanking you for the opportunity to revise our manuscript, we look forward to hearing your decision.

Yours sincerely,

Naroa

CP-2024-13 REVISION NOTES

COMMENTS BY THE EDITOR AND THE REVIEWERS, ALONG WITH THE AUTHORS' RESPONSES (HIGHLIGHTED IN COLOUR)

**Comments by the editor (Gerilyn Soreghan)**, and the authors' responses (highlighted in grey; same colour code for changes in the revised/marked manuscript)

This is an interesting manuscript on the origin and implications of Jurassic carbonate-marl successions, and is well-supported with a multi-proxy dataset and quantitative analyses. It is quite long; the authors should strive to shorten the text somewhat.

RESPONSE: AGREE. Section 5.2 of the manuscript has been significantly shortened by transferring former subsection 5.2.1 to an appendix. In addition, superfluous text of the original manuscript has been removed, revised version being considerably more concise. Consequently, section 5.2 in the revised version is 6% shorter than the original manuscript. However, following the editor's and reviewers' comments, some new information has been incorporated into the manuscript, mainly in the results section (e.g, new section 4.1.2), which has extended the revised version.

The authors have provided very thoughtful responses to the reviewer comments, and have indicated they will revise the manuscript in accordance with those comments, or have in some cases indicated and explained completely if they disagree. Once these changes are made-- which are a bit more than "minor" but not necessarily "major", the work should be accepted.

I do have a question about the authors' response to point #18 of reviewer #1; the authors state that "Limestones usually present higher magnetic susceptibility values than adjacent marls/shales." This is odd, because carbonate minerals are diamagnetic, and thus usually exhibit negative values.

RESPONSE: It is true that the magnetic susceptibility of hemipelagic deposits is commonly determined by paramagnetic phases (mostly detrital clays) and generally anticorrelates with carbonate content (e.g., Kodama and Hinnov, 2015). However, the opposite relationship was found in the Santiurde section. Based on this relationship and susceptibility-temperature (k-t) curves (Fig. S3), we consider that magnetite is the main driver of the MS signal in Santiurde (see lines 324-330 of the revised/marked manuscript), which is more abundant in limestones. According to Hunt et al. (1995), the average mass susceptibility of calcite, which is the main carbonate mineral in Santiurde, ranges from -0.3 to $1.4 \times 10^{-8}$ m$^3$/kg. The mass susceptibility of magnetite varies between 20000 and $110000 \times 10^{-8}$ m$^3$/kg, which is 4-6 orders of magnitude greater than that of calcite. Thus, a very low content of magnetite in limestones can reverse the low, or even negative (diamagnetic), mass susceptibility of the more abundant calcite.

Additionally, please see minor edits by line numbers below:

37— should be "diminished"

RESPONSE: AGREE. See line 39 of the revised/marked manuscript.

63— should this be "peri-Tethyan" ?

RESPONSE: AGREE. See line 70 of the revised/marked manuscript.

72— the aim IS to…

RESPONSE: AGREE. See line 79 of the revised/marked manuscript.

77— delete "firstly"

RESPONSE: AGREE. See line 84 of the revised/marked manuscript.

145— shouldn't this be "mass-normalized"? (Also elsewhere — e.g. 261)

RESPONSE: AGREE. See lines 167 and 322 of the revised/marked manuscript.

208— delete "weather" (also elsewhere, e.g. line 216)

RESPONSE: AGREE. Following the suggestion by reviewer 2 (Sietske Batenburg), we describe beds as either resistant or susceptible to weathering.

209— use "recessively weathering"

RESPONSE: PARTLY AGREE. Sentence rephrased in another way (see lines 232-235 of the revised/marked manuscript).

263— as noted above, this (higher MS in the limestones) seems strange to me, as carbonates are diamagnetic, so I don't understand how to explain this.

RESPONSE: See response to the comment above.

412— "were" deposite

RESPONSE: AGREE. See line 475 of the revised/marked manuscript.

413— delete "were" here

RESPONSE: AGREE. See line 476 of the revised/marked manuscript.

460— fix spelling to "rhythmites"— and in all other places used — do a global search for this and variations thereof, as it appears misspelled in many places.

RESPONSE: AGREE. The spelling of "rhythmites" has been revised throughout the manuscript.

483— replaced "avoided" with "prevented"

RESPONSE: AGREE. See line 552 of the revised/marked manuscript.

542— use "argument"

RESPONSE: Sentence removed.

553— delete "it can be concluded that" — also in 561

RESPONSE: AGREE. See line 626 of the revised/marked manuscript.

575— replace "no" with "not"

RESPONSE: AGREE. See line 648 of the revised/marked manuscript.

591— use "produced" instead of "originated" — also in 655, 975

RESPONSE: AGREE. See lines 665, 732 and 1066 of the revised/marked manuscript.

684-85— delete "it should be taken into account"

RESPONSE: AGREE. See line 763 of the revised/marked manuscript.

706, 746, 807, 831, 850, 857— typos

RESPONSE: AGREE. See lines 785, 827, 897, 921, 941 and 948 of the revised/marked manuscript.

829— grain size modes/distributions of the siliciclastic beds could help distinguish between these options. But I understand that that is probably a later step (ie, there is enough in this paper already).

RESPONSE: AGREE. The grain size of the terrigenous fraction could help to discriminate its fluvial or aeolian origin. However, this was beyond the scope of our study.

Dear editor and author,

This paper is a solid work on the climate control on a Lower Jurassic hemipelagic succession in the Basque-Cantabrian Basin that contain interesting approaches to understand factors controlling its accumulation. Data, interpretations and discussion are very well organized (although some parts are not balanced: see comment 20; and the discussion is quite long and complex). Without a doubt, the paper deserves to be published. However, concerning descriptions (and related interpretations and discussions) four main aspects require to be deeply explained:

- hemipelagic character of the successions (see manly comments 1, 6);
- significance of color (see comments 10, 16, 18) and MS data (see comments 17, 18);
- criteria for definition of couplets (precession cycles) and bundles (eccentricity cycle) (see comment 15);
- characterization of the black shale package as a whole (see comments 7, 13, 26).

Other changes are suggested in order to state clear some concepts and description

**Introduction**

1. Pelagic rhythmites are presented as one of the key sedimentary successions recording orbital controlled climate changes (first paragraph). However, the studied succession is hemipelagic. It would be interesting to include: 1st) a brief definition of the term hemipelagic in the context of the studied BCB; 2nd) a brief explanation (and references) on the role of orbital-induce climate variations on this particular kind of sediments, compared to the pelagic ones

RESPONSE: AGREE. See lines 54-67 (2nd point), and 104-107 and 1423-1425 (1st point) of the revised/marked manuscript.

**Geological setting**

2. Lines 83-84: "which connected the Boreal Sea with the southern Tethyan Ocean". Better: "which connected the Boreal Sea with the northwestern Tethyan Ocean".

RESPONSE: AGREE. See lines 91-92 of the revised/marked manuscript.

3. Line 87: "source area was located in the semiarid belt". What do you mean thin "source area", emerged land?, shallow platform carbonate source area? Please, explain better.

RESPONSE: AGREE. We refer to the emerged source area (see line 95 of the revised/marked manuscript).

4. Specify if the distribution of the humid and semi-arid zones was stable for the entire Early Jurassic

RESPONSE: AGREE. Clay minerals from the Early Jurassic Peritethyan area (Dera et al., 2009; Deconinck et al., 2020) results were congruent with independent approaches (e.g. Rees et al., 1999; Arias, 2007), supporting the identification of paleoclimatic belts during the Pliensbachian–Toarcian interval. Anyway, the study area, being close to the boundary between two latitudinal climatic zones, was especially sensitive to astronomically driven climate change. Periodic changes in orbital parameters generally force latitudinal displacements of this boundary (Martinez and Dera, 2015). Consequently, the study area could have suffered greater or lesser influence of the humid or arid zones during astronomical cycles. This information has been more clearly explained in lines 98-99 of the revised/marked manuscript.

5. Use in Fig. 1, Early/Lower Jurassic instead of Lias.

RESPONSE: AGREE. See revised version of figure 1.

6. Line 104: "Pliensbachian (192.9–184.2 Ma) hemipelagic successions of the BCB.". I suggest deleting the time duration: I suppose the studied succession has not been time-calibrated so accurately.

RESPONSE: AGREE. See line 110 of the revised/marked manuscript.

The sedimentary environment of the successions requires a deep explanation. Notice the term "outer ramp" appears for the first time in the discussion (line 778). See also lines 677-679 "restricted paleogeographic setting").

RESPONSE: AGREE. The sedimentary environment has been more precisely explained in lines 102-103 of the revised/marked manuscript.

Revise also lines 841-843 ("basins depleted in oxygen": be careful, it sounds like a circular reasoning).

RESPONSE: AGREE. See lines 931-934 of the revised/marked manuscript.

7. Line 106: use "packages of alternating black shales and limestones/marly limstones" instead of "black shale intervals". It is important to state clear these black shales do not include only shales but also intercalated limestone/marly limestones. I think the word Interval has a time connotation.

RESPONSE: PARTLY AGREE. The term "black shale interval" has been commonly used by previous authors for the studied deposits (e.g., Rosales et al., 2004, 2006; Quesada et al., 2005; and references therein) and we consider it appropriate (the term "interval", in addition to the time connotation, also refers to the space between objects, units, points or states). However, in lines 113-116 of the revised manuscript it is properly explained that the black shale intervals are packages of alternating black shale layers and limestones/marly limestone beds, which are separated from each other by decametric intervals devoid of black shale layers, in which only hemipelagic marls, marly limestones and limestones occur.

8. Lines 130-132: "and 1 km north-west of a coeval section studied by others at the train station in the same locality…with which a bed by-bed correlation can be readily carried out." This sentence is more appropriate for the discussion (see also comment 26). In any case, it requires a deep explanation of how this correlation was made, without (I suppose) lateral continuity of outcrops.

RESPONSE: AGREE. See lines 147-149 and 545-548 of the revised/marked manuscript and the revised version of supplementary figure 1. Bed by-bed correlation between separate and discontinuous outcrops was carried out on visual grounds, by the identification of key beds with distinctive sedimentary features (mainly lithology and thickness) and characteristic bed arrangements in the succession.

9. Lines 132-137: Please state clearer the location and thickness of the studied succession. As far I understand the studied succession is 22.5 m thick and includes: the uppermost 2.5 m of the Puerto Pozazal Formation and the lowermost 20 m of the Camino Formation (including the first x-thick black shale package of this unit). However, in line 140 "30.40 m thick" is mentioned.

RESPONSE: AGREE. We apologize for conveying misleading information in our original manuscript. As pointed out by the other reviewer (Sietske Batenburg), the thickness of the studied section was not clearly presented. We analysed a 30.4-m-thick section in the outcrop, but the

lowermost 7.9 m were excluded from the cyclostratigraphic analysis because of poor exposure. Thus, in our original manuscript the top of the stratigraphic log was located at 30.40 m, but it started at 7.9 m. This means that the studied succession is actually 22.5 m thick. In order to present this information more clearly, the bottom of the studied succession has been established at 0 m and the top at 22.5 m. See revised versions of figures 2, 4, 6, 7, 9, 10, 11, 14 and the supplementary material.

**Materials and methods**

10. The average color of samples is used for cyclostratigraphic (spectral) analysis. However, there is not any analysis to elucidate the sedimentary vs. diagenetic significance of this feature.

RESPONSE: DISAGREE. As shown in lines 316-321 of the revised/marked manuscript and in the revised version of figure S2, there is a great positive correlation between the colour and %$CaCO_3$ of rock powder samples from Santiurde. Consequently, rock powder colour can be considered a good representation of %$CaCO_3$. In fact, colour measurement of rocks, as an indicator of rock composition, is a relatively cheap, fast and non-destructive technique commonly used for cyclostratigraphic analysis (Olsen et al., 1999; Dinarès-Turell et al., 2003, 2018; Batenburg et al., 2014; Lauretano et al., 2015; Li et al., 2023; Martínez-Braceras et al., 2023; Wan and Wei, 2024). As with any other compositional proxy, the cyclostratigraphic analysis of colour data series can be carried out regardless of whether the rocks retain their original colour/composition or this was subsequently affected by diagenesis. In fact, the result of the cyclostratigraphic analysis will help elucidate whether the original sedimentary composition is retained: if an orbital forcing can be readily identified, this will imply that the succession retains the original (primary) sedimentary signal (as in our case study); if no orbital influence were deduced, this could imply that either the original succession was not orbitally forced or, alternatively, that the orbital signal was tainted by diagenesis. In our case study, the primary sedimentary origin of %$CaCO_3$ is widely discussed in section 5.1. Both physical (sedimentology, orbitally modulated bed arrangement, etc.) and geochemical (inorganic isotopes, major and trace element content, etc.) evidence corroborate that our calcareous rhythmites (as defined by their colour and %$CaCO_3$ content) responded to primary environmental variations and do not reflect diagenetic overprinting.

11. Thin sections are mentioned in results, but not included here.

RESPONSE: DISAGREE. The original manuscript mentioned that petrographic analysis of one sample per bed was carried out. In order to make things clearer (comment 12 about line 177), we specify that 19 samples were analysed in line 201 of the revised manuscript ..

12. Please, explain the lithology of the studied bundle and samples: line 167: fifty-seven samples, include also here the values of x samples/bed; line 177: central part of each bed, include here also the total number of samples.

RESPONSE: AGREE. See lines 190-191 and 201 of the revised/marked manuscript.

**Results**

13. Lines 209-210: Concerning lithological terms, "limestones or marly limestones" and "marls or shales". Do you have calcimetric analysis of the entire succession to differentiated these lithologies?. Concerning the term "shale", please see previous comment 7. The black shale package has to be presented.

RESPONSE: AGREE. The lithologies of the entire succession were defined on visual and sedimentological grounds. It is better explained in lines 232-246 of the revised manuscript (section 4.1.1) that this can be readily done in the field by taking into account rock colour,

hardness (expressed by weathering), internal lamination, and fossil content. As clearly stated in the manuscript, calcimetric analysis was only performed in the interval studied in detail, where bed composition was determined quantitatively. The calcimetric results confirm that the visual description of facies is accurate (revised version of Fig. 6). A presentation of the Black Shale interval studied herein is included in lines 262-266 of the revised manuscript.

14. Description of lithologies and texture. In Fig. 2 (log), marly limestones of limestones with different texture are not drawn. I suggest to draw them. Also state clear the description of each lithology separately (also limestones and marly limestones; do they have bioturbation?) and then compare their main differences.

RESPONSE: AGREE. See revised versions of figures 2 and 4. More accurate descriptions of the main lithologies and textures are now presented in section 4.1.1 of the revised manuscript (also see the response to the previous comment), including bioturbation.

15. Lines 236-244 on couplets and bundles. This paragraph has to be separated in a subsection. The criteria for differentiating couplets are unclear: why the couples marl/shale to limestone/marly limestone (and not at the contrary)?; the "lithological contrast" for bundles is also very unclear (see also comment 13 on carbonate content of the entire succession). Do you see significant features at the boundaries of couplets or bundles or any trends withing couplets or bundles?.

RESPONSE: AGREE. A new subsection "4.1.2 Bed arrangement" has been added in the revised manuscript. The criteria used for the definition of couplets and bundles is now more clearly explained. As both couplets and bundles are cyclic arrangements of beds, they do not have objective boundaries with significant features. Thus, it is irrelevant whether couplets contain marls/shales below and limestones/marly limestones above, or vice versa, providing that the criterion is coherent throughout the succession. Bundles also show a symmetrical vertical trend in the arrangement of their component couplets. As defined herein, the lithological contrast of the beds that make up successive couplets increases progressively from the bottom to the middle part of the bundles (marl/marly limestone couplets at the bottom of the bundles, shale/limestone couplets in their middle parts), and then gradually decreases again (bundles ending up with marl/marly limestone couplets at their tops). Thus, bed boundaries are sharper in the middle part of bundles than at bundle boundaries, most likely due to the greater lithological contrast between successive beds in the former. Otherwise, neither couplet boundaries nor bundle boundaries show any significant features.

16. Color trends: lines 244-258 "The variations in colour values are more significant in the central couplets of bundles than at bundle boundaries. This suggests that, as shown in previous studies… colour values are representative of the carbonate content of the samples.". See previous comment 15 on "lithological contrast" for bundles (not well explained" and also comment 10 (significance of color). To use the similar trend in color and carbonate content in C35 to C44 as supporting criterion, it is necessary to discuss there was not a diagenetic imprint in both color and carbonate content.

RESPONSE: AGREE. The manuscript has been revised accordingly; see responses to comments 10, 13, 14 and 15.

17. Did you perform analysis of susceptibility-temperature (k-t) curves to know the type and abundance of magnetic minerals? The following sentence is not clear (as far I understand you interpret the presence of ferromagnetic minerals indirectly): Lines 264 "The MS of hemipelagic deposits is commonly determined by their paramagnetic components (mostly detrital clays; Kodama and Hinnov, 2015). However, in Santiurde this parameter does not show a great correlation with colour (r: 0.48, p<0.001, all section; Fig. S1) or calcium carbonate (r: 0.36,

p<0.001, between C35 and C44; Fig. S1). Therefore, the Santiurde relationship suggests that the MS signal is more likely controlled by ferromagnetic minerals, such as magnetite (Fig. S2)." Revise also lines 750-755.

RESPONSE: UNCERTAIN ABOUT THIS COMMENT. As stated in the original manuscript, susceptibility-temperature (k-t) curves were obtained, and the result of a representative sample presented in the figure S3 in the revised supplementary material. The thermomagnetic curve confirms the presence of magnetite, which is thought to be the main MS driver.

18. Spectral analysis of MS data (lines 283-285). MS data do not correlate with color and carbonate content; however, their spectral analysis corroborate the results of the spectra analysis of color. Please, explain this apparent contradiction.

RESPONSE: AGREE. An explanation is given below and has been incorporated into the caption revised version of figure S4 in the revised supplementary material. Limestones usually present higher magnetic susceptibility values than adjacent marls/shales. However, the MS data series displays a greater dispersion and a spikier appearance than the colour and %CaCO$_3$ series, which very likely explains the low correlation coefficient between the MS data series and the colour and %CaCO$_3$ data series (figure S2 in the revised supplementary material). As explained in the response to comment 17, the MS signal is mainly carried by magnetite content, which could be either detrital in origin or related to postdepositional changes in redox state. The influence of early diagenetic processes, such as partial replacement of pyrite with iron oxides at more oxygenated conditions, might explain the high variability of the MS curve. Notwithstanding the potential flaws of the MS data series, the spectral analysis shows that it records a significant periodicity with an average thickness equivalent to that of precession couplets. Despite being less prominent, cycles correlatable with those attributed to obliquity(?), short eccentricity (bundles) and long eccentricity in the colour spectral analysis series can also be identified in the MS spectra.

19. Lines 310-311. "In general, %CaCO3 fluctuates in line with lithology, limestones and marly limestones (average: 66.36%) being richer than marls and shales (average: 34.86%). What do you mean? In fact, carbonate content is the criterion to differentiate these lithologies.

RESPONSE: AGREE. The lithology of the entire succession was defined on visual grounds. The high-resolution calcimetric analysis of Bundle 9 (C35-C44 interval) corroborates the visual lithological identification. See lines 373-375 of the revised/marked manuscript.

20. In 4.2. Detailed analysis of Bundle 9 (C35-C44 interval), pure descriptions are included in 4.2.1 to 4.2.4; however, 4.2.5 and 4.2.6 contain interpretation/discussion of the results, including the interpretation of oxic/anoxic conditions of the different lithologies (without any reference to the other results). This imbalance should be corrected.

RESPONSE: DISAGREE. In sub-section 4.2.5, the enrichment factors of several elements and some palaeoceanographic indices are calculated and presented. In order to understand why these (and not other) indices and elements are analysed, we consider it necessary to explain their palaeoenvironmental meaning and significance (with references to others' works). However, the specific results of the elemental enrichment factors and the paleoceanographic indices from Santiurde are not interpreted in this sub-section (this is done later in section 5), only their general trends are described.

Similarly, only the results of a factor analysis are presented in sub-section 4.2.6. There was only one interpretation at the end of this subsection in the original manuscript (referring to orbital forcing), which has been modified in the revised manuscript in line with the reviewer's comment (see lines 523-524 of the revised/marked manuscript). There are no other interpretations of the results obtained in our study in this subsection. Other statements that may resemble interpretations

(the palaeoenvironmental meaning of the most representative elements or group of variables extracted from the factor analysis) are, again, simple reminders of the basic concepts introduced in the preceding subsection, which intend to help the reader follow our line of reasoning.

**Discussion**

21. Line 459. "Origin of inorganic sedimentary fluctuations". I suggest deleting "inorganic". This term is obscure.

RESPONSE: AGREE. See line 526 of the revised/marked manuscript.

22. Lines 470-472 (secondary cements..), line 474 (bed geometry): these descriptions should be explained also in Results.

RESPONSE: AGREE. The manuscript has been revised accordingly, transferring several characteristics of the succession mentioned in the discussion into section 4.1.1. (Sedimentology and petrography of the general Santiurde section, lines 232-235 and 244-246 of the revised/marked manuscript.).

23. Lines 476-478. "Quite the opposite, the characteristics of the beds are continuous for more than 1 km between the Santiurde motorway and railway sections". See comment 8.

RESPONSE: AGREE. This sentence has been modified in line with reviewer's comment 8 (see above).

24. Lines 485-487: "In general, the diagenetic characteristics observed in the Santiurde rhytmites are typical of processes related to organic matter decay during burial (Rosales et al., 2001). This sentence is not informative. Please explain in which way.

RESPONSE: AGREE. The sentence in lines 555-557 of the revised/marked manuscript has been deleted.

25. Lines 488-493 about periodicities. Do you have data on the time span of the studied succession to compare with your results? I would be interesting to know how many cycles are then represented in the entire succession and BS package.

RESPONSE: AGREE. We have estimated the duration of the studied interval and of the Black shale interval 1 by counting orbital cycles. However, we will not include this at the position pointed out by the reviewer (in the discussion about the primary or diagenetic origin of the rhythmite), but in section 5.3 about orbitally modulated environmental changes (see lines 868-872 of the revised/marked manuscript.).

26. The discussion lacks a proper explanation of the BS package as a whole (how many precession or eccentricity cycles includes, what short- and long-term factors controlled its accumulation.

RESPONSE: PARTLY AGREE. The duration of the BS package has been estimated based on the number of orbital cycles, which will be added in the revised manuscript (response to comment 25). The orbitally modulated environmental factors that controlled the fluctuating sedimentation when the Black Shale interval was being accumulated are widely discussed in section 5.3. However, the factors the determined the formation of the entire Black Shale interval cannot be elucidated with the data available in this study. As stated by Rosales et al. (2006), the Pliensbachian Black Shale intervals of the BCB accumulated during second order sea level rises.

Regards,
Beatriz Bádenas

**Comments by Reviewer 2 (Sietske Batenburg)**, and the authors' responses (highlighted in blue; same colour code for changes in the revised/marked manuscript)

Review Martinez-Braceras, CP, 2024

The manuscript on 'Orbitally forced environmental changes during the accumulation of a Pliensbachian (Lower Jurassic) black shale in northern Iberia' by Martinez-Braceras and co-authors investigates factors driving sedimentary rhythms in a Pliensbachian black shale interval. This is a timely approach, as sedimentary rhythms in Mesozoic successions are commonly used to construct astronomically tuned time scales, although the exact mechanisms driving lithological alternations (especially on the precessional scale) are often insufficiently understood. The authors present a multi-proxy study to discuss a suite of processes in detail, shedding light on the periodic nature of regional anoxia that resulted in the deposition of organic matter. This is a very thorough study and merits publication in Climate of the Past if some minor points can be addressed.

As the stratigraphic interval has been studied previously in the same region, it would be relevant to report any independent age information. This would include biostratigraphic, magnetostratigraphic and chemostratigraphic events, and if available (correlation to) radioisotopic ages. If no age information is available, the authors should make it clear that the interpretation of orbital forcing of the sedimentary rhythms is based solely on the cycle hierarchy.

RESPONSE: AGREE. The presentation of the chronostratigraphic data available for the studied interval has been improved (see lines 153-159 and 563-565 of the revised/marked manuscript), which is based on ammonite zones (Braga et al., 1988) and calcareous nannofossil zones (Fraguas et al., 2015). Unfortunately, the biostratigraphic age data does not provide the resolution needed to assess accurately the chronology of this relatively short succession at astronomical timescales.

The photograph of the section (Fig 2) shows very clear banding patterns, with individual lithological alternations varying in intensity and showing grouping in bundles. The time series analyses of the colour signal show periodicities at 6.6, 1.67, 1 and 0.37 m, where the 'intermediate' periodicities have some peculiarities. The periodicity at 1.6 m is not very strongly present in the time series analysis, whereas it is prominent in the log and view of the section. The periodicity at 1 m seems a bit different from what would be expected with a cycle hierarchy of eccentricity-modulated precession and obliquity (20:5:2:1). The longest periodicity that is strongly present in the spectral results likely reflects the influence of 405 kyr cycles, but its expression in the section is not clear. It would be good if the authors could comment on the reasons for the seeming discrepancies in the lithological patterns and the spectral analysis result.

RESPONSE: AGREE. It is true that calcareous couplets of 32-42 cm and bundles of approximately 1.65 m constitute the clearest bed arrangement in the outcrop, which correspond to the expression of precession (20 kyr) and short (100 kyr) eccentricity cycles, respectively. However, long eccentricity cycles (405 kyr) are also expressed in the section by the alternation of 3.3-m-thick intervals in which two successive short eccentricity bundles are clearly recorded (e.g., B9 and B10), and 3.3-m-thick intervals in which another two short eccentricity bundles are not so clearly defined (e.g., the underlying B7-B8 and the overlying B11-B12; see lines 304-309 of the revised/marked manuscript). The former intervals are interpreted as long eccentricity minima, the latter as maxima. We have added a new figure (Fig. S1A), where the record of 405 kyr cycles can be readily appreciated. As the physical expression of short eccentricity bundles is subdued at long eccentricity maxima, the power of the former is relatively weak in the time series analysis. This explains the relatively low intensity of the intermediate, 1.6-m-thick periodicity in the spectra.

The time series analyses of the colour signal show a significant 1-m-thick cyclicity, which could not be identified visually in the outcrop. Based on an average duration of 20 kyr for each

precession-driven couplet, this intermediate periodicity would represent 53 kyr, which could correspond to the 52.8 kyr term of obliquity (p+S6). This 1-m-thick cyclicity is not recorded in the MS spectra, but a less significant periodicity of 65 cm is identified. Based on the average duration of 20 kyr for each calcareous couplet, this intermediate MS periodicity would represent 35 kyr. The mean duration estimated from both proxies is 44 kyr, suggesting that they might be the result of obliquity. However, as the results of the two datasets are not fully coherent, these periodicities (marked as O? in Fig. 4) were not further considered in our discussion about the orbitally modulated environmental evolution of the area.

A related point is that the periodicities detected through time series analyses are consistently shorter than those observed. The number of interpreted bundles (14) in a 31 m interval suggest that the imprint of short eccentricity actually resulted in a 2.2 m cycle rather than a 1.6 m one. The 6.6 m periodicity has a much stronger peak in the spectrum but based on the number of individual alternations (62), it is only present 3 times in the studied section, and has a length of approx. 10 m rather than 6.6 m. The individual alternations have an average thickness of 31/62=0.5 m rather than 0.37 m. I do not understand the origin of this discrepancy. It has been observed that the highest amplitude cycles may have higher sedimentation rates (in the absence of dissolution) and these thicker couplets may dominate the time series analyses results. But here, the time series results indicate shorter periodicities. I would recommend the authors to evaluate other power spectra methods, to see whether these give similar results, and to report on the imposed settings more elaborately. Also, it would be interesting to generate a power spectrum for $CaCO_3$ in the studied high-resolution interval, to see whether the statistically identified periodicity driving the limestone marl alternations corresponds to the observed thickness of the alternations.

RESPONSE: AGREE. We apologize for conveying misleading information in our original manuscript. As pointed out by the other reviewer (Beatriz Badenas), the thickness of the studied section was not clearly presented. We analysed a 30.4-m-thick section in the outcrop, but the lowermost 7.9 m were excluded from the cyclostratigraphic analysis because of poor exposure. Thus, in our original manuscript the top of the stratigraphic log was located at 30.40 m, but it started at 7.9 m. This means that the studied succession is actually 22.5 m thick. In order to present this information more clearly, the bottom of the studied succession has been established at 0 m and the top at 22.5 m in the revised manuscript (see revised versions of figures2, 4, 6, 7, 9, 10, 11, 14 and the supplementary material).

Taking into account the thickness of 22.5 m of the studied section, the periodicities detected in the time series analyses are consistent with those observed in the succession. The 13.4 bundles identified in the 22.5 m thick succession present an average thickness of 1.68 m, which matches the 1.67 m periodicity deduced from the spectral analyses. Similarly, the 62 calcareous couplets display an average length of 36 cm, which is practically identical to the 37 cm peak and the average of the 32-42 cm band identified in the spectral analyses. Additionally, the 2π-MTM power spectrum of the $CaCO_3$ data of the bundle (B9) studied in detail (see figure below) identifies a significant peak between 31-42 cm (mean value of 36.7 cm), which also corresponds to the thickness of the limestone-marl couplets.

[Figure]

It would be good to indicate how couplets and eccentricity bundles are defined here precisely, along the lines of: 'The term couplet, as used here, refers to a lithological alternation, consisting of a resistant limestone bed with a more weathered marl or shale bed, starting at the base of the marl or shale. These couplets vary in their amount of lithological contrast between the marl/shale and the limestone. The variations in lithological contrast result in a grouping into bundles of five (four to six) couplets, counting from the base of the lightest coloured marls, reflecting the least lithological contrast with their bounding limestones.'

RESPONSE: AGREE. Following also the other reviewer's comment, a specific section (4.1.2. Bed arrangement) has been added in the revised manuscript.

The L/M ratio being close to 1 is taken as indication that carbonate productivity and dilution varied hand in hand. Besides this ratio, it would be interesting to plot the thickness of the couplets and the thickness of the individual beds, to see whether for example thicker couplets coincide with thicker limestones or not.

RESPONSE: DISAGREE. The interpretation that carbonate productivity and dilution varied together is not only based on the L/M thickness ratio, but also on other geochemical characteristics obtained from the interval studied in detail (bundle B9 at 12.4-15.95 m). Such a comprehensive dataset is not available for the entire section (0-22.5 m), which means that it is not possible to deduce the variations in environmental conditions over time based solely on L/M thickness ratios. However, following the reviewer's suggestion, we calculated the L/M ration of the entire succession (0-22.5 m). The average L/M ratio of 1.08 obtained from the couplets of the entire Black Shale interval (C10 to C45 at 2.5-16.4 m) is similar to that obtained in bundle B9. The figure below shows the thickness and L/M ratio of all the couplets. There is not a clear or repetitive trend between the thickness of couplets and their L/M ratio. Consequently, we consider that this information does not contribute significantly to the main scope of our study.

[Figure]

A range of geochemical methods is applied to carefully investigate the factors controlling the production and preservation of organic matter. Changes in P-EF seem to suggest elevated productivity in the dark levels, but this is contrasted by δ15Norg, δ13Corg and Ba-EF, which are explained to suggest lower productivity. I wonder if the authors can comment on whether, instead of increased productivity, enhanced preservation would be sufficient to explain the observed patterns.

RESPONSE: AGREE. The fact that changes in the preservation of organic matter constituted the main factor that controlled the $C_{org}$ content is now more clearly explained in lines 846-854 of the revised/marked manuscript.

The $\delta^{13}C$ changes are addressed in many parts of the manuscript, and perhaps the readability would benefit from grouping all information about δ13C together, or a paragraph summarizing it.

RESPONSE: DISAGREE. It would be rather difficult to concentrate the discussion about $\delta^{13}C_{carb}$ data in one single section, because it is used to asses both the diagenetic overprinting and the orbitally modulated environmental changes (sections 5.1 and 5.3). Moreover, $\delta^{13}C_{carb}$ results were not more significant than other geochemical or mineralogical proxies (none of which is discussed in specific sections) for the development of the cyclic sedimentation model. In this regard, the only exception is the content in organic matter (specifically addressed in section 5.2), but this is due to the fact that the great organic matter content is the main feature that characterizes the Basque-Cantabrian Lower Jurassic successions and why we selected this interval for our study.

Minor points

L 21: change 'involved processes' to 'processes involved'
RESPONSE: AGREE. See line 21 of the revised/marked manuscript.

L 22: change 'The study' to 'This study'
RESPONSE: AGREE. See line 22 of the revised/marked manuscript.

L 23: change 'black shales' to 'black shale intervals', change 'revealed' to 'reveals'
RESPONSE: AGREE. See lines 23-24 of the revised/marked manuscript.

L 25: the phrase 'with the prevalence of precession, short eccentricity and long eccentricity cycles' could be replaced by something a like: 'and were likely driven by eccentricity-modulated precession' to be more precise
RESPONSE: AGREE. See lines 25-27 of the revised/marked manuscript.

L 32: the comma has to be deleted to understand what the active verbs are in the sentence
RESPONSE: AGREE. See line 33 of the revised/marked manuscript.

L 34: change waters to water
RESPONSE: AGREE. See line 35 and others throughout the revised/marked manuscript.

L 36: change maximum to maximal
RESPONSE: AGREE. See line 37 of the revised/marked manuscript.

L 37: typo in diminished

RESPONSE: AGREE. See line 39 of the revised/marked manuscript.

L 38: delete seawater

RESPONSE: AGREE. See line 40 of the revised/marked manuscript.

L 39: add and before contributed

RESPONSE: PARTLY AGREE. This sentence will be rephrased in another way in order to improve its meaning (see lines 39-40 of the revised/marked manuscript).

L 40: change exportation to export

RESPONSE: AGREE. See line 41 of the revised/marked manuscript.

L 43: change seawaters to water

RESPONSE: AGREE. See line 44 of the revised/marked manuscript.

L 46: change orbital to orbitally

RESPONSE: AGREE. See line 47 of the revised/marked manuscript.

L 50: change few to a few

RESPONSE: AGREE. See line 52 of the revised/marked manuscript.

L 51: add 'and temporal' after latitudinal

RESPONSE: AGREE. See line 53 of the revised/marked manuscript.

L 56: move 'erode the seabed' and 'or' to before 'interrupt'

RESPONSE: AGREE. See lines 59 of the revised/marked manuscript.

L 57: delete ', sedimentation'

RESPONSE: AGREE. See lines 61-62 of the revised/marked manuscript.

L 80: delete on

RESPONSE: AGREE. See line 87 of the revised/marked manuscript.

L 82: replace Armorican by 'the Armorican Massif'

RESPONSE: AGREE. See line 89 of the revised/marked manuscript.

L 83: replace being part of by 'within'

RESPONSE: AGREE. See line 90 of the revised/marked manuscript.

L 99: delete was

RESPONSE: AGREE. See line 101 of the revised/marked manuscript.

L 106: Here and in other occasion: I recommend avoiding the abbreviation BS which is in English is commonly used to refer to bullshit.

RESPONSE: AGREE. BS will be replaced with BSI throughout the text to refer to "Black shale interval".

L 111: replace United Kindom with 'the United Kingdom'

RESPONSE: AGREE. See lines 120-121 of the revised/marked manuscript.

L 116: put in before inland

RESPONSE: AGREE. See line 125 of the revised/marked manuscript.

L 120: replace on by of

RESPONSE: AGREE. See line 129 of the revised/marked manuscript.

L 208: weather resistant and weather recessive does not sound correct. You could delete 'weather' or you could explain that the beds are either resistant or susceptible to weathering.

RESPONSE: AGREE. See lines 232-235 of the revised/marked manuscript.

L 216: as L 208

RESPONSE: AGREE. See lines 247-248 of the revised/marked manuscript.

L 218: add the before marls

RESPONSE: AGREE. See line 249 of the revised/marked manuscript.

L 220: add and before trace

RESPONSE: DISAGREE. The information in brackets is a list of characteristics, separated by semicolons.

L 234: delete weather (2x)

RESPONSE: AGREE. See line 272 of the revised/marked manuscript.

L 277: replace which peaks at by with a main periodicity of

RESPONSE: AGREE. See line 337 of the revised/marked manuscript.

L 288: to help the reader, please mention the width of the filters in the main text, expressed as periodicities. Consider explaining why the bandwidths of the two filters are very different (half of the centre frequency vs one fourth of the centre frequency).

RESPONSE: AGREE. See lines 346-350 of the revised/marked manuscript. We agree that there was no coherence between the bandwidths of both filter outputs. Consequently, a new filter output has been extracted for the intermediate frequency and included in the new Figure 5, with a bandwidth close to one fourth of the centre frequency (similar to that used in the filter of the short periodicity).

L 293: I suggest to replace the word chronostratigraphy by cyclostratigraphic interpretation. Ideally, an integrated chronostratigraphy would include information from bioevents, magnetostratigraphy, chemostratigraphy, radioisotopic dating, etc.

RESPONSE: PARTLY AGREE. The chronostratigraphic information refers to the Jamesoni biozone of the Pliensbachian stage (obtained from Quesada et al., 2005, and Rosales et al., 2006), depicted on the left of the stratigraphic log. In Figure 4 we do not present our cyclostratigraphic interpretation of the Santiurde section, but only the results of our spectral analysis. In order to clarify this misunderstanding, we have added the reference for the chronostratigraphic data in line 355 of the revised/marked manuscript.

L 299: add cycles after m, replace corresponds by correspond

RESPONSE: AGREE. See lines 359-361 of the revised/marked manuscript.

L 311: add 'in CaCO3' after richer

RESPONSE: AGREE. See line 374 of the revised/marked manuscript.

L 313: delete counterpart

RESPONSE: AGREE. See line 377 of the revised/marked manuscript.

L 340: add with before maximum

RESPONSE: AGREE. See line 403 of the revised/marked manuscript.

L 375: replace 'the amplitude of the oscillations' by 'amplitude of variability'

RESPONSE: AGREE. See lines 438-439 of the revised/marked manuscript.

L 422: replace seawater by water, and 'concentration was' by 'concentrations were'

RESPONSE: AGREE. See lines 485-486 of the revised/marked manuscript.

L 434: replace whose by which, add of before their

RESPONSE: AGREE. See line 500 of the revised/marked manuscript.

L 557: add with before that

RESPONSE: AGREE. See line 630 of the revised/marked manuscript.

L 565: replace records by record

RESPONSE: AGREE. See line 639 of the revised/marked manuscript.

L 567: replace alternation by alternations

RESPONSE: AGREE. See line 641 of the revised/marked manuscript.

L 577: replace if by when

RESPONSE: AGREE. See line 650 of the revised/marked manuscript.

L 606: add than after higher

RESPONSE: AGREE. See line 1201 of the revised/marked manuscript.

L 625: replace indurate by indurated

RESPONSE: AGREE. See line 1220 of the revised/marked manuscript.

L 632: replace indurate by indurated

RESPONSE: AGREE. See line 1225 of the revised/marked manuscript.

L 655: replace originate by lead to

RESPONSE: AGREE. See line 732 of the revised/marked manuscript.

L 660: add strength of before biological pump

RESPONSE: AGREE. See line 737 of the revised/marked manuscript.

L 664: replace distortions by alterations

RESPONSE: AGREE. See line 742 of the revised/marked manuscript.

L 672: replace come by coincide

RESPONSE: AGREE. See line 749 of the revised/marked manuscript.

L 676: replace 'and greater OM with relatively higher' by 'and more OM with a relatively higher'

RESPONSE: AGREE. See line 754 of the revised/marked manuscript.

L 681: the $P_{Ef}$ record actually does not always have its maxima in black shales. Perhaps mention the P concentrations themselves to strengthen the observation.

RESPONSE: AGREE. See line 759 of the revised/marked manuscript.

L 706: typo in would

RESPONSE: AGREE. See line 785 of the revised/marked manuscript.

L 721: replace sea bottom by either sea floor or bottom water

RESPONSE: AGREE. See line 798-799 of the revised/marked manuscript.

L 766: replace bottom by floor

RESPONSE: AGREE. See line 848 of the revised/marked manuscript.

L 769: replace waters by water

RESPONSE: AGREE. See line 853 of the revised/marked manuscript.

L 785: typo in oxygenation

RESPONSE: Sentence removed.

L 835: replace are no evidences by is no evidence

RESPONSE: AGREE. See line 925 of the revised/marked manuscript.

L 844: replace depth by depths

RESPONSE: AGREE. See line 935 of the revised/marked manuscript.

L 862: typo in diagenetic

RESPONSE: AGREE. See line 953 of the revised/marked manuscript.

L 885: add the before OM

RESPONSE: AGREE. See line 977 of the revised/marked manuscript.

L 1076: replace supplies by supply

RESPONSE: AGREE. See line 1168 of the revised/marked manuscript.

L 1090: delete sea

RESPONSE: AGREE. See line 1183 of the revised/marked manuscript.

L 1096: typo in significant

RESPONSE: AGREE. See line 1189 of the revised/marked manuscript.

Figure 4: replace 'Relief in the outcrop' by 'weathering profile'. Consider reverting the colour axis back, so that the peaks coincide with protruding beds and are more easily compared with the log.

RESPONSE: AGREE. See the revised version of figure 4.

Figure 13: This is an excellent summary of your findings and the different orbital configurations are explained well. Instead of Ti/Al and Si/Al, I recommend using the enrichment factors that you use in the text and other figures. As the role of productivity is not so well constrained, I recommend using a question mark after the claim of increased productivity. Similarly, you could consider including only low/high OM preservation in the text within the figure (rather than including the transport)

RESPONSE: AGREE. See the revised version of figure 13.

**Cited literature (not included in the manuscript):**

Arias, C.: Pliensbachian–Toarcian ostracod biogeography in NW Europe: evidence for water mass structure evolution, Palaeogeogr. Palaeoclimatol. Palaeoecol., 251(3-4), 398-421, https://doi.org/10.1016/j.palaeo.2007.04.014, 2007.

Batenburg, S.J., Gale, A.S., Sprovieri, M., Hilgen, F.J., Thibault, N., Boussaha, M. and Orue-Etxebarria, X.: An astronomical time scale for the Maastrichtian based on the Zumaia and Sopelana sections (Basque country, northern Spain), J. Geol. Soc., 171(2), 165-180, 2014.

Dinarès-Turell, J., Baceta, J. I., Pujalte, V., Orue-Etxebarria, X., Bernaola, G. and Lorito, S.: Untangling the Palaeocene climatic rhythm: an astronomically calibrated Early Palaeocene magnetostratigraphy and biostratigraphy at Zumaia (Basque basin, northern Spain), Earth Planet. Sci. Lett., 216(4), 483-500, 2003.

Hunt, C. P., Moskowitz, B. M. and Banerjee, S. K.: Magnetic properties of rocks and minerals. Rock physics and phase relations: A handbook of physical constants, 3, 189-204, 1995.

Kodama, K.P. and Hinnov, L.A.: Rock magnetic cyclostratigraphy, John Wiley & Sons, Oxford, UK, 165 pp., ISBN 978-1-118-56128-7, 2015.

Lauretano, V., Littler, K., Polling, M., Zachos, J. C. and Lourens, L. J.: Frequency, magnitude and character of hyperthermal events at the onset of the Early Eocene Climatic Optimum. Clim. Past, 11(10), 1313-1324, 2015.

Olsen, P. E. and Kent, D. V.: Long-period Milankovitch cycles from the Late Triassic and Early Jurassic of eastern North America and their implications for the calibration of the Early Mesozoic time–scale and the long–term behaviour of the planets. Phil. Trans. R. Soc. A, 357(1757), 1761-1786, 1999.

Rees, P. M., Alfred, M. Z. and Paul J.V.: Jurassic Phytogeography and Climates: New Data and Model Comparisons, in: Warm Climates in Earth History, edited by Huber, B. T., Macleod, K. G. and Wing, S. L., Cambridge, Cambridge University Press, UK, 297–318, https://doi.org/10.1017/CBO9780511564512.011, 1999.

Wan, J. and Wei, Z.: Unveiling chromaticity and Milankovitch cycles in sedimentary rocks via unmanned aerial vehicle photogrammetry, Newsl. Stratigr., 57 (2), 235-256, DOI: 10.1127/nos/2024/0815, 2024.